



# Evaluating the effect of nutrient redistribution by animals on the phosphorus cycle of lowland Amazonia

Corina Buendía[1,2], Axel Kleidon[1], Stefano Manzoni[3,4], Bjorn Reu[5], and Amilcare Porporato[6]

[1]Biospheric Theory and Modelling group, Max Planck Institute for Biogeochemistry, Hans-Knöll Str. 10, 07745 Jena, Germany
[2]Corporación Colombiana de Investigación Agropecuaria (Corpoica), km 32 vía al mar, Rionegro, Santander, Colombia
[3]Department of Physical Geography, Stockholm University, 10691 Stockholm, Sweden
[4]Bolin Center for Climate Research, Stockholm University, 10691 Stockholm, Sweden
[5]Escuela de Biologia, Universidad Industrial de Santander, 6344000 Bucaramanga, Santander, Colombia
[6]Civil and Environmental Engineering Department, Duke University, Durham, NC 27708, USA

*Correspondence to:* Corina Buendía (coribuend@gmail.com)

**Abstract.** Amazonian ecosystems are of global importance for the climate system and biodiversity. Phosphorus (P) is suggested to be a limited nutrient in many Amazonian ecosystems because soils are ancient, highly weathered, and nutrient depleted. Recently, it has been suggested that large herbivores may play a major role in the Amazon nutrient cycle. Here, we develop this hypothesis further and show how the spatial redistribution of P from rivers to land, across different ecosystems and between
sub-basins may even sustain the Amazonian P cycle, and this not only by means of large herbivores, which supposedly were present previously to human arrival to the contient, but by different animal foraging strategies that exist today. To do so, we introduce a simple mathematical framework, which synthesizes the major processes of the Amazonian P cycle and allows for quantifying their relative contribution to the P budget. With this model we use sensitivity analyses to demonstrate how animals can affect the P cycle. Our findings suggest the importance of the interweaved Amazonian ecology, including fish migrations,
different flooding and soil moisture regimes as well as the role of different animal foraging strategies for a sustainable P cycling in these ecosystems.

## 1  Introduction

The Amazon basin counts as one of the most biodiverse regions on Earth, including highly productive ecosystems, essential to the regulation of the global climate system. It covers about 7 million km$^2$: thirteen percent of which is covered by the Andes,
while the rest is characterized by relatively flat topography. Over millions of years, the topographical gradient has resulted in a gradient of soil fertility from young and nutrient-rich Andean soils to ancient and highly weathered soils in the central and lower Amazon basin (Hoorn et al., 2010). Consequently, rivers originating in the Andes (called "white water rivers") transport nutrient-rich sediments, whereas rivers originating in the lowlands tend to be nutrient poor ("black water rivers" if they carry organic acids and "clear water rivers" if they do not). Because of nutrient transport by the white water rivers, an even steeper
nutrient availability gradient exists between the river floodplains covering about 30% of the basin (Junk et al., 2011b), and the *terra firme* ecosystems, which do not receive nutrients by seasonal floods. Ecosystems that are seasonally flooded by white





waters are traditionally called *Várzea* and are characterized by a high primary productivity and tree diversity (Wittmann et al., 2006), as compared to the ecosystems seasonally flooded by clear or black waters, which are called *Igapó* (Fig. 1). Not only differences in nutrients, but also differences in precipitation result in diversity among Amazon ecosystems, such as seasonally flooded rain forests, *terra firme* rain forests, dry forests, wetlands (Pantanal), and tropical savannas (Cerrado). In the lower part

of the Amazon where water is less limiting, these ecosystems sustain a particularly high productivity and diversity of life forms (Gentry, 1992; Antonelli and Sanmartín, 2011).

Phosphorus (P) is a crucial element for life, providing structure to RNA and DNA and with a key function in energy transfer and storage (ATP and ADP). In general weathering is the main source of P to terrestrial ecosystems. Theories on pedogenesis suggest that under humid climates and slow tectonic uplift, rock weathering becomes negligible preventing input of "fresh"

phosphorous to the biosphere (Chadwick et al., 1999; Walker and Syers, 1976; Wardle, 2004; Wardle et al., 2009; Crews et al., 1995). Under such conditions, without major disturbances (e.g. glaciation resetting soil development), P availability and with it net primary productivity decreases, leading to the so-called *retrogressive phase* (Wardle, 2004; Wardle et al., 2009) or *terminal steady-state* (Walker and Syers, 1976). However, despite their 100 million years old soils (Hoorn et al., 2010), some ecosystems in lower Amazonian basin are among the most diverse and productive (Gentry, 1992). This raises the question as

to what prevents Amazon ecosystems from falling into a *retrogressive phase* or *terminal steady-state*?

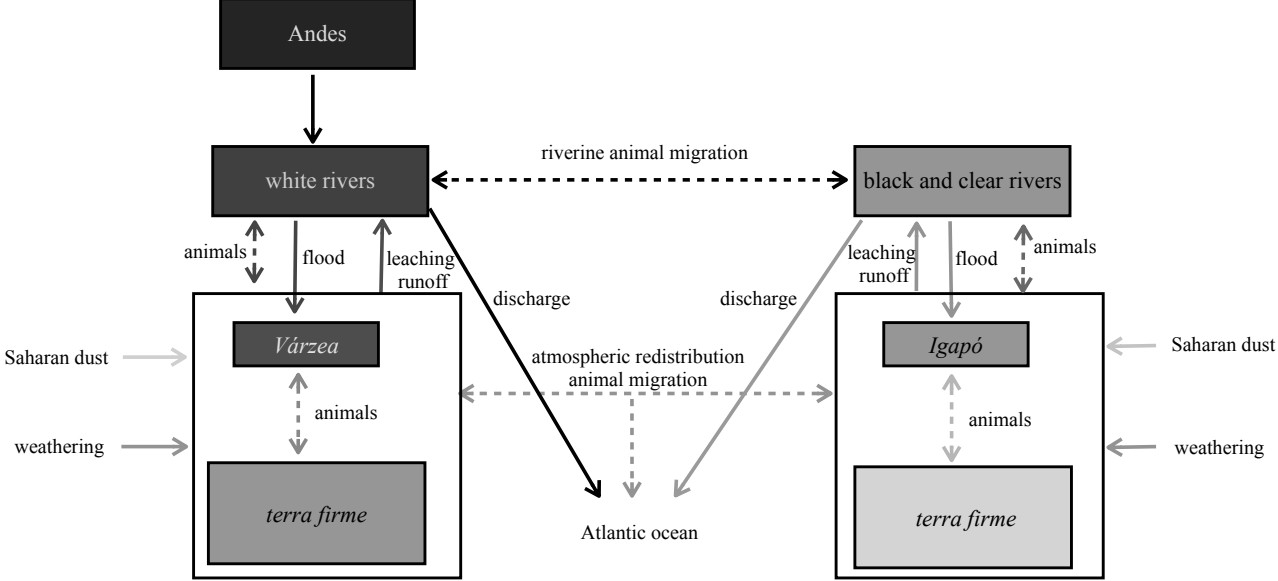

**Figure 1.** Conceptual diagram of processes transferring P across ecosystems in the Amazon basin. The boxes represent the major eco-regions of Amazonia and arrows represent the fluxes between them. The grey tone represents the P content of these eco-regions from dark (high P content) to light tones (lower P content).

Here we shortly review what we know about the P budget of lowland Amazonian ecosystems as illustrated in Figure 1. We start by discussing the inputs from the bedrock by weathering and then how likely or not is that deocclusion of P in clays could





serve as a long term P source. Later, we discuss the exogenous inputs; first those transported by the atmosphere and then those transported by the river and how they could be reaching flooded and *terra firme* ecosystems.

Weathering in the central Amazon basin was estimated to be about 75 $\mathrm{g\,P\,ha^{-1}a^{-1}}$ based on data taken at the mouth of Rio Negro river, which is an important tributary of the Amazon river draining only the lowlands (Gardner, 1990). This measurement partly contradicts, what we have learned from soil chronosecuences like Hawaii Islands and Franz Joseph glacier retrogretion (Walker and Syers, 1976; Wardle, 2004; Wardle et al., 2009). The Amazon basin experiences continental isostatic rebound, where the slow erosion rates are compensated by slow uplift and weathering of new material (Porder et al., 2007; Buendía et al., 2010). However, because bedrock can be as deep as 100 m it is not clear whether or not P released by weathering can reach the terrestrial biotic cycle (Gardner, 1990). Nevertheless, as dissolved P reaches the rivers by ground-water flow, it can be used by freshwater ecosystems.

High amounts of P are found in occluded forms in the old soils of central Amazon. Life has evolved energetical-costly mechanisms, like cluster roots and mycorrhizal associations to make some of this P available to vegetation (Lambers et al., 2008). In a previous paper (Buendía et al., 2014) this possibility was explored by formulating a model that in a simple but explicit way, accounts for physical and chemical weathering, secondary mineral formation, P occlution, and P deocclution at a carbon cost. Our modeling study and logical reasoning suggests that because Amazonian soils are very old and the pool of occluded P is finite, it cannot support the ecosystems on the long run. Nevertheless, it can act as a reserve of P for the ecosystem.

Dust originating from African deserts carry P to the Amazon basin, but this contribution is highly uncertain, spanning two orders of magnitude, from 4.8 $\mathrm{g\,P\,ha^{-1}a^{-1}}$(Mahowald et al., 2005), to 11-47 $\mathrm{g\,P\,ha^{-1}a^{-1}}$ (Swap et al., 1992), and 125 - 426.47 $\mathrm{g\,P\,ha^{-1}a^{-1}}$ (Bristow et al., 2010). Dust comprises only about 7 to 17 % of atmospheric deposition, while a much larger fraction is composed of biogenic particles (83-90 %) originating from the Amazon basin itself. Ashes originating from fires might also contribute to the internal P redistribution in some regions (Artaxo and Hansson, 1994; Mahowald et al., 2005; Pauliquevis et al., 2012). Biogenic particles, such as pollen, spores, bacteria, algae, protozoa, fungi, and leaf fragments are generated by the forest and to a great extend may also terminate in the forest again. However, at the same time about 19 $\mathrm{g\,P\,ha^{-1}a^{-1}}$ are deposited into the ocean via atmospheric transport (Mahowald et al., 2005), where it becomes an important nutrient source to Atlantic marine ecosystems near the continent. Overall, for the terrestrial ecosystems of the Amazon basin the atmosphere could even act as P sink, rather than a net P source.

Junk et al. (2011a) estimated that 30% the Amazon basin comply with international criteria for wetland definition. Rivers seasonaly flood different areas with this providing sediments and P inputs, which is therefore referred to as 'flood-pulse' concept (Junk, 1997; Junk et al., 1989). Despite the fact that flood pulses are well documented, (Junk, 1997; Junk et al., 1989, 2011a), it is difficult to quantify the magnitude of this P flux as it varies depending on the flooding intensity, type of sendiments and matter transported, soils type, and the functional composition of the *Várzea* and *Igapó* ecosystems.

In addition to abiotic driven P fluxes, migratory animals, like fish, caimans, turtles, and birds migrate on a seasonal basis between the Andean-influenced white waters to P-deficient lowland black and clear waters. Animal migration thus results in a redistribution of nutrients within and across different sub-basins. This connection between sub-basins is well studied for some



catfish species and have been shown to be significant but difficult to quantify for the clear and black water sub-basins (McClain and Naiman, 2008; Barthem and Goulding, 1997).

Within the sub-basins, P may be transported from aquatic to terrestrial ecosystems by animals feeding on riverine food sources, like the jaguar, the giant otter and fishing birds. For example, an adult giant otter (*Pteronura brasiliensis*) consumes

about 3 kg of fish per day (Carter and Rosas, 1997). Assuming fish dry weight is 20 % and P content of fish at about $1.1 - 4.5\%$ (Sterner and Elser, 2002), an adult otter could transfer about 6.6 - 27 g P per day to terrestrial ecosystems (2409 - 9855 $\mathrm{g\,P\,a^{-1}}$). Using the population density reported for Suriname, of 1.2 individuals per $\mathrm{km^{-2}}$ (Duplaix et al., 2008), giant otters could contribute about 28 - 118 $\mathrm{g\,P\,ha^{-1}a^{-1}}$. Although this species is an endangered species, currently, there are no density estimates available. This movement can be complemented by terrestrial animals including soil fauna, insects and mammals that

frequently utilize both seasonally flooded and *terra firme* habitats. Figure 2 presents an illustration of the different animals that can be seen during the dry (lower half) and wet (upper half) seasons in a black water rivers, as perceived by indigenous people of the Rio negro sub-basin. The movement between habitats further enhances the P-redistribution potential at finer spatial scales. In other words, animal movement within the Amazon basin generate a transport mechanism for P. Animal activity therefore triggers a diffusion-like transport of P from relatively rich hotspots to poorer areas, typically against the gradients of physical

flow processes. For example, in a study of a woolly monkey (*Lagothrix lagothricha lugens*) population in northwest-Amazon, Stevenson and Guzmán-Caro (2010) showed that dual habitat used by this population could import about 1-4 $\mathrm{g\,P\,a^{-1}}$ through seed dispersal. A model proposed and illustrated how mega fauna before the Pleistocene extinction could have acted as such redistribution agent (Doughty et al., 2013).

However, it has not been demonstrated that entire ecosystems involving different trophic levels can achieve this in a similar

manner. Here we show, using a simple mathematical framework, that P from seasonally flooded ecosystems can be redistributed from the floodplains enriching P-poor *terra firme* ecosystems. Because the model accounts for different transport pathways including different animal foraging strategies (piscivory, herbivory and detritivory) in a minimal parameterization, it allows for evaluating the relative importance of different redistribution mechanisms.

Due to the high complexity of food webs and biotic interactions, it is impossible to consider all P fluxes driven by animal

migration and movement and the differences between the different ecosystems. Therefore, it is our objective to quantitatively evaluate the importance of the different redistribution mechanisms using a model synthesizing the major processes of P cycling, including P redistribution by animals (Fig. 1). As a first approximation, rather than focusing on a single mode of transport (as in Doughty et al. (2013)), we consider two foraging strategies of consumer animals - herbivory and detritivory. Both strategies allow the spatial redistribution of biomass and P. To assess the importance of biomass consumption by animals on

P redistribution under different environmental settings, we parameterized our model for three contrasting Amazonian lowland sub-basins, each subdivided into seasonally flooded (P-richer *Várzea* and P-poorer *Igapó*) and *terra firme* areas: (1) the Rio Negro sub-basin, a lowland black-water tributary of the Amazon River which is generally poor in P, (2) the Caquetá-Japurá sub-basin, a white water river rich in Andean sediments and hence relatively richer in P, and (3) the Cerrado, a dry tropical savanna ecosystem seasonally flooded by clear waters (which are poor in P). By doing so, we account for the main environmental





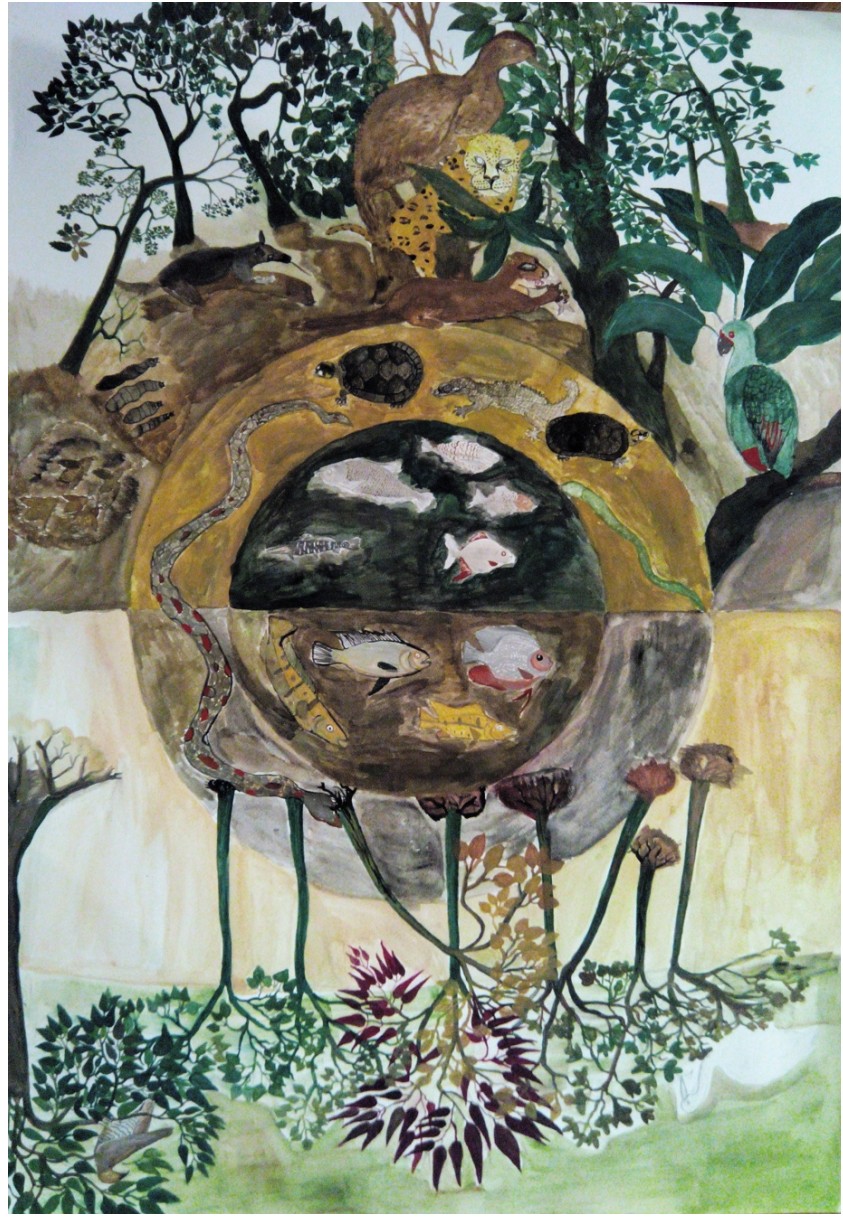

**Figure 2.** Painting illustrating the fauna that can be observed near the Miriti, a black water tributary of the Caquetá river. The upper half-sphere represents the dry season, when most of the terrestrial animal are present in this area. The lower half-sphere represents rainy season, when animals move deep into the *terra firme* forest or to the head waters. With the beginning of the dry season animals feed on what remained from the flood and is the time of the year when turtles and caimans lie their eggs, which are often consumed by terrestrial animals. This original water painting was illustrated by Marcela and Johana Yucuna, indigenous of the Yucuna ethnic group from the Mirití region (Caquetá, Colombia)



variability affecting our model results, such as differences in P load of the flooding river (white-waters vs. black- and clear-waters) and differences in soil moisture regime affecting the P losses from the ecosystem (dry vs. humid).

## 2 Methods

### 2.1 Modeling framework

The model includes a 'local' P cycling module (based on Buendía et al. (2010)), a description of plant- and detritus-animal interactions, and an animal-driven P redistribution mechanism between seasonally flooded and *terra firme* ecosystems. The 'local' module consists of six ordinary differential equations representing the dynamics of weatherable material, secondary minerals, occluded P, P in available forms ($P_d$), P in vegetation biomass ($P_v$), and P in soil and litter biomass ($P_o$). The model uses annually averaged soil moisture content, and the processes are interpreted at the annual time scale. Since animal

dynamics are much faster than weathering and occlusion, it is safe to assume that animal pools are in quasi-equilibrium, whereas weathered P, occluded P, and P in secondary minerals are at steady state, thereby reducing the number of equations from six to three. We parameterized the system for seasonally flooded ($F$) and *terra firme* (or upland $U$) ecosystems, coupling these two ecosystems through two biotic P fluxes representing the effect of herbivores ($H$) and detritivores ($D$) (refer to Fig. 3). This distinction determines from which biomass pool animals feed (i.e., from live or dead biomass, respectively). A summary

and description of symbols and parameters is given in Tables 1 and 2. Our model assumes that herbivory and detritivory redistribute P within and between both ecosystems. This approximation could in the future be relaxed to account for the fact that animals tend to consume nutrient-rich foliage and detritus, which is more abundantly available in the seasonally flooded forest (Andersen et al., 2004) and hence a greater proportion of it may be transferred to *terra firme* ecosystems (i.e. directional P redistribution).

As consider in this model two transport processes control P redistribution by the terrestrial animal food webs, one that is supported by live vegetation biomass (herbivory) and one that is supported by litter and soil organic matter (detritivory). Herbivory here is represented as a consumption fraction $k_H$ of standing vegetation biomass ($P_{vE}$) and represents the foraging strategy of, for example, monkeys, birds, and leaf cutter ants together with their supported food web (see Fig. 3). Detritivory is modeled as a consumption fraction $k_D$ of soil and litter organic biomass ($P_{oE}$) and represents the foraging strategy of species

such as termites, soil fauna and their food webs.

  In other words, accounting for the source areas, herbivores consume $k_H(A_U P_{vU} + A_F P_{vF})$, whereas detritivores consume $k_D(A_U P_{oU} + A_F P_{oF})$. These fluxes are then returned to the available P and detritus P compartments in the seasonally flooded ($F$) and *terra firme* ecosystems ($U$). Each ecosystem receives a fraction of the total consumption equal to its fractional area ($A_F$ and $A_U$, respectively). Of this input, animals mineralize a fraction $k_{HM}$ (transferred to $P_{dE}$), whereas the

remaining fraction $(1 - k_{HM})$ is transferred to $P_{oE}$. Assuming that animals do not export P outside of the sub-basin and that they do not preferentially transport P towards *terra firmes* or seasonally flooded ecosystems the averaged losses equal the inputs. Based on these assumptions, the non-mineralized organic inputs to both ecosystems are mathematically expressed as: $AI_{voE} = (1 - k_{HM})k_H(A_U P_{vU} + A_F P_{vF})$ and $AI_{ooE} = (1 - k_{DM})k_D(A_U P_{oU} + A_F P_{oF})$ for herbivory and detritivory,



**Table 1.** Description of symbols. Subscript $E$ may stand for either *terra firme* (replaced by $U$), or seasonally flooded ecosystem (replaced by $F$).

| Type | Symbol | Mathematical description | Description | Units |
|------|--------|--------------------------|-------------|-------|
| Pools | $P_{vE}$ | | phosphorus in vegetation | $\mathrm{g\,P\,m^{-2}}$ |
| | $P_{oE}$ | | phosphorus in soil biomass | $\mathrm{g\,P\,m^{-2}}$ |
| | $P_{dE}$ | | phosphorus in soil solution | $\mathrm{g\,P\,m^{-2}}$ |
| Fluxes | $F_{dcE}$ | $k_c P_{dE}$ | phosphorus occlusion | $\mathrm{g\,P\,m^{-1}\,a^{-1}}$ |
| | $F_{dvE}$ | $P_{dE}\frac{\eta s_E}{nZ_r s_E}$ | phosphorus uptake by vegetation | $\mathrm{g\,P\,m^{-2}\,a^{-1}}$ |
| | $F_{voE}$ | $P_{vE}k_v$ | phosphorus losses from vegetation | $\mathrm{g\,P\,m^{-2}\,a^{-1}}$ |
| | $F_{odE}$ | $P_{oE}k_d\frac{s_E T}{20}$ | phosphorus mineralization | $\mathrm{g\,P\,m^{-2}\,a^{-1}}$ |
| Losses | $O_{oE}$ | $P_{oE}(k_f + k_r k_l s_E^c)$ | phosphorus in organic form | $\mathrm{g\,P\,m^{-2}\,a^{-1}}$ |
| | $O_{dE}$ | $P_{dE}\frac{k_l s_E^c}{nZ_r s_E}$ | phosphorus in soil solution | $\mathrm{g\,P\,m^{-2}\,a^{-1}}$ |
| Animal fluxes | $AOo_E$ | $k_D P_{oE}$ | detritivores animal consumption $P_{oE}$ of | $\mathrm{g\,P\,m^{-2}\,a^{-1}}$ |
| | $AO_{vE}$ | $k_H P_{vE}$ | herbivores animal consumption of $P_{vE}$ | $\mathrm{g\,P\,m^{-2}\,a^{-1}}$ |
| | $AI_{odE}$ | $k_{DM}k_D(A_F P_{oF} + A_U P_{oU})$ | detritivores mineralized inputs $P_{vE}$ | $\mathrm{g\,P\,m^{-2}\,a^{-1}}$ |
| | $AI_{ooE}$ | $(1 - k_{DM})k_D(A_F P_{oF} + A_U P_{oU})$ | detritivores inputs $P_{vE}$ | $\mathrm{g\,P\,m^{-2}\,a^{-1}}$ |
| | $AI_{vdE}$ | $k_{HM}k_H(A_F P_{vF} + A_U P_{vU})$ | herbivores mineralized input $P_{vE}$ | $\mathrm{g\,P\,m^{-2}\,a^{-1}}$ |
| | $AI_{voE}$ | $(1 - k_{HM})k_H(A_F P_{vF} + A_U P_{vU})$ | herbivores organic inputs $P_{vE}$ | $\mathrm{g\,P\,m^{-2}\,a^{-1}}$ |

respectively. The inputs in mineralized forms $P_{dE}$ are described as $AI_{vdE} = k_{HM}k_H(A_U P_{vU} + A_F P_{vF})$ for herbivory and $AI_{odE} = k_{DM}k_D(A_U P_{oU} + A_F P_{oF})$ for detritivory. Because animals have limited assimilation efficiency we assume that a fraction $k_{HM}$ of this animal P input $I_{aF}$ goes to the dissolved pool $P_d$ and the rest to the organic pool $P_o$. Therefore we defined: $I_{aoF} = (1 - k_{HM})I_{aF}$ and $I_{adF} = I_{aF}k_{HM}$.

Note that, if herbivore consumption is set to 10 %, the model assumes that 10% of both the biomass of both seasonally flooded ($F$) and the *terra firme* ecosystems ($U$) is consumed. Therefore, the magnitude of the redistribution linearly depends on the P stocks in each ecosystem. Animal population dynamics are effectively neglected here (e.g., the model would not properly describe herbivore outbreaks that defoliate large areas), although other approaches to modeling herbivory include an animal pool and employ nonlinear consumption kinetics (e.g., de Mazancourt and Schwartz (2010); Doughty et al. (2013)).

The advantage of our minimal approach is that it does not require any parameter except the consumption rate of vegetation and detritus, for which we show sensitivity analyses, and the partitioning of animal P to organic and available pools, which does not play a major role at steady state.

Complementary to his function is terrestrial animals that transport P from river to flooded ecosystems, for example giant otters, fishing birds and humans are represented here as annual fluxes of P to flooded areas as the animals will probably use

*terra firme* areas close to the rivers. It is important to emphasize that we do not model animal movement *per se*, but rather we account for the fact that animal movement between ecosystem types redistributes P because of differences in P availability of



**Table 2.** Description of model parameters.

| Type | Parameter | Description | Value | Units | Reference |
|---|---|---|---|---|---|
| common | $\eta$ | maximum transpiration rate | 5 | mm day$^{-1}$ | Porporato et al. (2003) |
| | $T$ | temperature | 25 | Celsius | |
| | c | exponent of runoff leakage function | 3 | unit-less | Buendía et al. (2010) |
| | $Z_r$ | effective soil depth | 1 | m | Buendía et al. (2010) |
| | $n$ | porosity | 0.4 | dimensionless | Buendía et al. (2010) |
| | $k_c$ | phosphorus occlusion rate | 0.00001 | m$^2$a$^{-1}$g$^{-1}$ | re-calibrated |
| | $k_e$ | wind and gravitational driven losses | 0.00001 | a$^{-1}$ | Buendía et al. (2010) |
| | $k_l$ | runoff/leakage rate at saturation | 0.1 | a$^{-1}$ | Buendía et al. (2010) |
| | $k_d$ | mineralization rate | 0.19 | a$^{-1}$ | Buendía et al. (2010) |
| | $k_v$ | litter fall rate | 0.20075 | a$^{-1}$ | Buendía et al. (2010) |
| | $k_r$ | losses regulation rate | 0.002 | a$^{-1}$ | Buendía et al. (2010) |
| | $k_u$ | active uptake by vegetation | 10 | dimensionless | Buendía et al. (2010) |
| | $k_f$ | wind, animal, fire losses rate | 0.0001 | a$^{-1}$ | Buendía et al. (2010) |
| | $I_w$ | weathering | 80 | g P ha$^{-1}$a$^{-1}$ | steady-state solution |
| | $I_d$ | atmospheric deposition of dust | 5, 11-47 | g P ha$^{-1}$a$^{-1}$ | Swap et al. (1992) |

the various pools and differences in spatial extent of the ecosystem types. It is the net effect of all these transport mechanisms, mediated by the whole food web rather than driven by single animal functional types, that causes a net transfer of P.

### 2.1.1 Balance equations

The following equations represent the P balance of a generic ecosystem (subscript $E$). The specific equations for *terra firme* and seasonally flooded ecosystems are obtained by replacing subscript $E$ by $U$ for upland and $F$ for flooded, respectively. The parameters used for seasonally flooded and *terra firme* ecosystems are the same, with the exception of yearly averaged soil moisture $s_E$ and their spatial extent ($A_U$ and $A_F$). Vegetation obtains P from available forms in the soil ($P_{dE}$) through water uptake and symbiotic organisms ($F_{dvE}$); losses are due to herbivory ($AO_{vE}$) and litterfall ($F_{voE}$, see Fig. 3),

$$\frac{dP_{vE}}{dt} = F_{dvE} - F_{voE} - AO_{vE}, \tag{1}$$

Soil and litter organic biomass ($P_{oE}$) increase due to litterfall ($F_{voE}$) from the same ecosystem as well as from the connected ecosystem due to the contribution of herbivores ($AI_{voE}$), detritivores ($AI_{ooE}$), and terrestrial animals feeding on riverine food sources (also refered as terrestrial piscivores) that transport P to flooded ecosystems ($I_{aoF}$). P release due to mineralization of soil organic matter is a function of soil moisture and temperature. Detritivores also induce mineralization and redistribution of $P_{oE}$ through the flux $AO_{oE}$. Accordingly, the mass balance equation for organic matter P reads,

$$\frac{dP_{oE}}{dt} = I_{aoF} + F_{voE} - F_{odE} - O_{oE} - AO_{oE} + AI_{voE} + AI_{ooE}. \tag{2}$$

P in available forms ($P_d$) receives inputs from atmospheric dust deposition ($I_d$), weathering ($I_w$), flooding in seasonally flooded ecosystems ($I_{fF}$), terrestrial piscivore imports to flooded ecosystems ($I_{adF}$), mineralization of $P_{oE}$ ($F_{odE}$), and mineralization



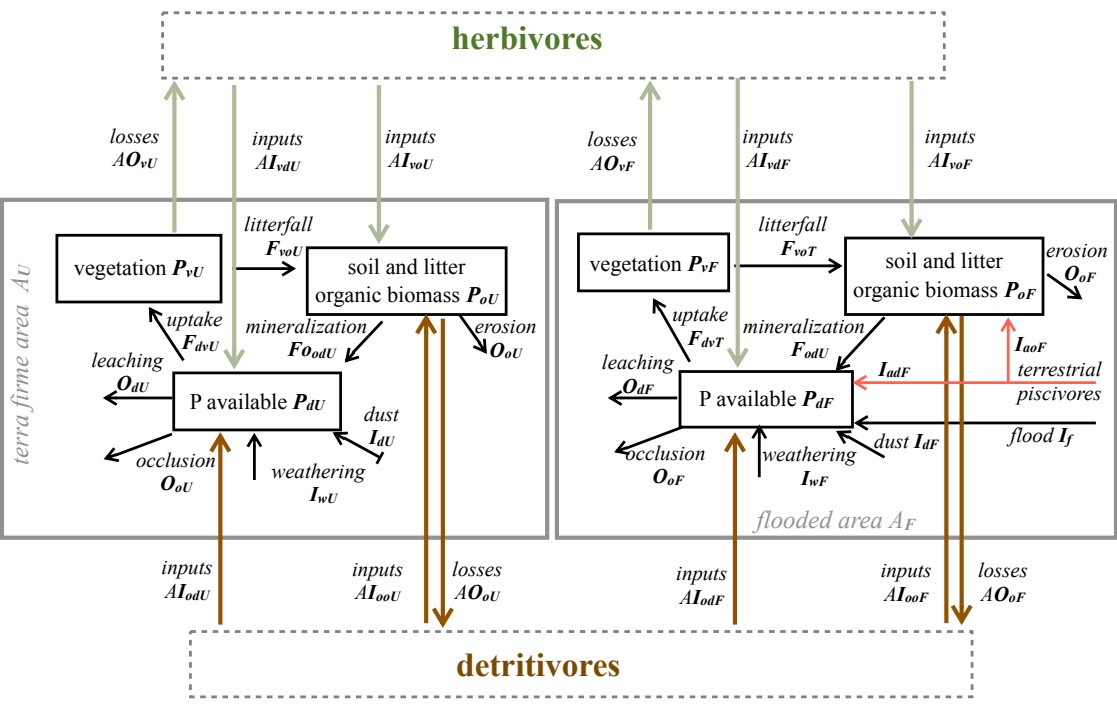

**Figure 3.** Diagram of model structure representing P fluxes and pools. Black arrows represent the P fluxes among pools representing the basin dynamics of the P cycle and color arrows represent the P fluxes due to animal consumption and dual habitat use: green arrows for herbivory fluxes and brown arrows for detritivory fluxes, red arrow for terrestrial piscivores.

and redistribution through animals ($AI_{odE} + AI_{vdE}$). Losses are driven mainly by runoff and leaching $O_{dE}$, vegetation uptake $F_{dvE}$, and occlusion $F_{dcE}$, so that the mass balance equation for available P can be written as,

$$\frac{dP_d}{dt} = I_d + I_w + I_{fF} + I_{adF} + F_{odE} - [F_{dvE} + F_{dcE} + O_{dE}] + AI_{odE} + AI_{vdE}. \tag{3}$$

Fluxes accounting for animal activity are described here, while other fluxes follow the model by Buendía et al. (2010) (Table
5    1), where they are described in detail.

## 2.2    Model parameterization and inputs

### 2.2.1    Weathering inputs

For the weathering flux, we assumed an average molar P concentration in the bedrock of 75 mol P m$^{-3}$ (Porder et al., 2007), and a tectonic uplift rate for the lowland basin of 0.0057 mm a$^{-1}$ (Kronberg et al., 1979). We assumed that uplift rates are the
10    same for the whole basin, and that only 60% of the material in the rock is weatherable. The steady-state solution corresponds to a weathering flux of about 80 g P ha$^{-1}$a$^{-1}$. Furthermore, we assume that the P from weathering is available for plants.





### 2.2.2 Atmospheric input

While P in gaseous phase forms is not common, the atmosphere can transport P carrying particles. As it has been already discuss in the introduction that deposition of Saharan dust has been found to contribute to the P budget of the Amazon basin and we chose to work with 5 $\mathrm{g\,P\,ha^{-1}a^{-1}}$. The atmospheric deposition of biogenic particles, which accounts for the highest

percentage of atmospheric deposition (about 80 % in the Amazon), should not be considered as a system input, but rather as a sub-basin recycling process (Mahowald et al., 2005). It should be noted that the term identified in our model as detritivory could also be considered as a sub-basin recycling process. Hence, the production of biogenic particles by forests has a similar effect as detritivore redistribution within the basin.

### 2.2.3 Flooding inputs

Andean soils are weathered and eroded at very high rates compared to the lowlands, leading to differences in the nutrient loads of rivers originating from the Andes and the lowlands, as well as for the seasonally flooded ecosystems associated to these river types. Seasonally flooded ecosystems are classified into *Várzea* and *Igapó*. The origin of the rivers and their transported sediments, the soil moisture regime, and duration of flooding play an important role for the P input to seasonally flooded areas. To constrain the range of parameter values in the sensitivity analyses, we estimate the maximum possible P input based on

weathering estimates of sub-basins of the Amazon basin. The calculation is based on the assumption that the P cycle is at steady state, i.e., P transport by rivers out of the basin equals the weathering rate as it was explained in the introduction.

Gardner (1990) estimated the P weathering rate for the Amazon and Rio Negro basins to be 457 and 242 $\mathrm{mol\,P\,km^{-2}a^{-1}}$ respectively. Assuming that the lowland of the Amazon have a similar weathering rates the Rio Negro sub- basin, and taking into account that lowlands occupy about 87% of the whole basin, their contribution to the total is about $242*0.87 = 210.54$

$\mathrm{mol\,P\,km^{-1}a^{-1}}$. Hence, the Andes contribute with the remaining $457-210 = 246$ $\mathrm{mol\,P\,km^{-2}a^{-1}}$.

Since the Andes cover 13% of the total basin, the in-situ weathering must be around 1895.84 $\mathrm{mol\,P\,km^{-2}a^{-1}}$. Because our goal here is to define an upper limit for the sensitivity analysis, we assume that all the P from Andean weathering is deposited through flooding to the seasonally-flooded areas. These areas occupy 30 % of the drained area, so that the total amount of P that is deposited amounts to $1895/0.3 = 6316$ mol P $\mathrm{mol\,P\,km^{-1}a^{-1}}$ (i.e. 1957 $\mathrm{g\,P\,ha^{-1}a^{-1}}$). Following similar calculations, for

*Igapó* ecosystems the deposition rate is estimated as 700 mol P $\mathrm{mol\,P\,km^{-2}a^{-1}}$ (i.e. 217 $\mathrm{g\,P\,ha^{-1}a^{-1}}$).

### 2.2.4 Parameterization of animal dynamics

The parameterization of animal dynamics requires only a few parameters. P from animal turnover and excreta is assumed to be transferred equally to either available P and soil organic matter P (i.e., $k_{HM} = k_{DM} = 0.5$). Altering this assumption did not significantly affect the results.

For the animal P flux from rivers to the flooded areas, simulations with three different P inputs were run, with values of 0, 72 and 242 $\mathrm{g\,P\,ha^{-1}a^{-1}}$. The first one simulates a scenario with no animals, the second simulates a scenario in which P transfer





**Table 3.** Description of model parameters that are site specific.

| Type | Parameter | Description | Value | Units | Reference |
|---|---|---|---|---|---|
| site specific | $A_U$ | fraction of land covered with *terra firme* ecosystems | 0.7 | dimensionless | Junk et al. (2011b) |
| | $A_F$ | fraction of land covered with flooded ecosystems | 0.3 | dimensionless | Junk et al. (2011b) |
| | $s_U$ | yearly averaged soil moisture of *terra firme* ecosystems | 0.2 - 0.6 | dimensionless | variable |
| | $s_F$ | yearly averaged soil moisture seasonally flooded ecosystems | 0.7 | dimensionless | chosen |
| | $I_{f_W}$ | inputs by seasonal flooding to *Várzea* ecosystems | 1566 | $\mathrm{g\,P\,ha^{-1}a^{-1}}$ | variable to 1957 |
| | $I_{f_B}$ | inputs by seasonal flooding to *Igapó* ecosystems | 196 | $\mathrm{g\,P\,ha^{-1}a^{-1}}$ | variable to 250 |
| | $I_{aF}$ | inputs from river to land by animals | 0, 72, 242 | $\mathrm{g\,P\,ha^{-1}a^{-1}}$ | variable |
| animal driven | $k_D$ | litter and soil organic matter consumption by detritivores | 0-0.3 | $\mathrm{a^{-1}}$ | variable |
| | $k_H$ | vegetation consumption by herbivores | 0-0.1 | $\mathrm{a^{-1}}$ | variable |
| | $k_{DM}$ | mineralization fraction due to herbivory | 0.5 | $\mathrm{a^{-1}}$ | chosen |
| | $k_{HM}$ | mineralization fraction due to herbivores | 0.5 | $\mathrm{a^{-1}}$ | chosen |

is mainly due to giant otter (*Pteronura brasiliensis*), with an associated P flux of 72 $\mathrm{g\,P\,ha^{-1}a^{-1}}$ assuming population density reported for Suriname, and the last one 242 $\mathrm{g\,P\,ha^{-1}a^{-1}}$ simulates a scenario in which otters and other animals contribute.

## 2.3 Scenarios for Amazonian sub-basins

To assess the importance of biomass consumption by animals on P redistribution under different environmental settings, we
parameterized our model for three contrasting Amazonian lowland sub-basins, each subdivided into seasonally flooded (P-richer *Várzea* and P-poorer *Igapó*) and *terra firme* areas: (1) the Rio Negro sub-basin, a lowland black-water tributary of the Amazon River which is generally poor in P, (2) the Caquetá-Japurá sub-basin, a white water river rich in Andean sediments and hence relatively richer in P, and (3) the Cerrado, a dry tropical savanna ecosystem seasonally flooded by clear waters (which are poor in P). By doing so, we account for the main environmental variability affecting our model results, such as differences
in P load of the flooding river (white-waters vs. black- and clear-waters) and differences in soil moisture regime affecting the P losses from the ecosystem (dry vs. humid).

    At the regional scale, the average fluxes are calculated with the assumption that 30% of the terrestrial area is seasonally flooded and the 70% is *terra firme* (non-flooded) (Junk et al. 2011). We run the model for *terra firme* ecosystems ($U$) using yearly averaged relative soil water content of 0.35 for the Cerrado (Runyan and D'Odorico, 2012), and 0.6 for the *terra firme*
part of the Caquetá-Japurá and Rio Negro sub-basins. Furthermore, for all seasonally flooded areas in the three sub-basins ($F$), we assumed yearly averaged soil water content of 0.7.





## 2.4 Simulation setup

The solution of the system of six ordinary differential equations was obtained using the deSolve package in R (Soetaert et al., 2010). The model approaches steady-state at around 7000 simulation years. Since the initial state of the system is not known, we use here only the steady-state solutions for our results.

## 3 Results

P redistribution by herbivores and detritivores has a different effect on the P status of vegetation (4 vs. 5), which also depends on the soil moisture regime of the sub-basin (Rio Negro vs. Cerrados vs. Caquetá-Japurá) and the ecosystem type (seasonally flooded vs. terra firme).

Humid or dry *terra firme* ecosystems have less P in biomass than their associated to *várzea* ecosystems, which is expected
due to the high P inputs due to flooding. In turn, humid *terra firme* ecosystems have less P than their associated *Igapó* ecosystems (flooded by P-poor water), while dry *terra firme* ecosystems have more P in biomass than their associated *Igapó* ecosystems.

P transfer by animals from river to land enhances P availability in flooded areas, and can change the previously described result, when combined with herbivory and detritivory as shown in the sensitivity analysis in Fig. 4. Notably, animal contribution
to P redistribution has an optimum at intermediate values of herbivore consumption. This optimum is caused by the combination of two contrasting effects; moderate herbivory allows P transport to *terra firme* ecosystems, where P can be stored due to reduced losses (i.e. less leaching), whereas excessive herbivore consumption increases P in the dissolved P pool, which is prone to leaching.

We find that a rate of herbivore consumption of 2% of P in living biomass maximizes the P status of the *terra firme* ecosys-
tem, while at a rate of 1% the P status of the whole sub-basin reaches its maximum. This difference of 1% originates from the P gains of the *terra firme* ecosystem, but also takes into account the losses of the seasonally flooded ecosystem that occupies 30% of the sub-basins area. For ecosystems influenced by nutrient-poor waters the P gradient between seasonally flooded and *terra firme* area is small and hence small herbivore consumption rates of <1% increase the P status of the *terra firme* ecosystems. Higher rates lead to a decrease of P in vegetation at the sub-basin scale due to the losses from the dissolved pool under humid
soil moisture regime. In contrast, in the simulation for the Cerrado without animal transfer of P from river to land, herbivory and detritivory negatively affect the P balance of the *terra firme* areas, since P is scare and redistribution causes more losses than gains. As a results of these processes, *terra firme* vegetation of the Cerrado has higher P (5 $\mathrm{g\,P\,m^{-2}}$) than the associated vegetation of the *Igapó* (4 $\mathrm{g\,P\,m^{-2}}$). Finally, the *terra firme* ecosystem of the Rio Negro sub-basin has the lowest amounts of P in vegetation (between 3 and 9 $\mathrm{g\,P\,m^{-2}}$, depending on the inputs). Overall, the sub-basin that has more P in vegetation is
the Caquetá-Japurá, followed by the Cerrado, and Rio Negro with the lowest.

In Figures 4 and 5 we also tested the role of P transport by piscivores from the rivers to the flooded areas, from which it may be transported further inland by detritivores and herbivores as described above. In all sub-basins, piscivores activity (solid



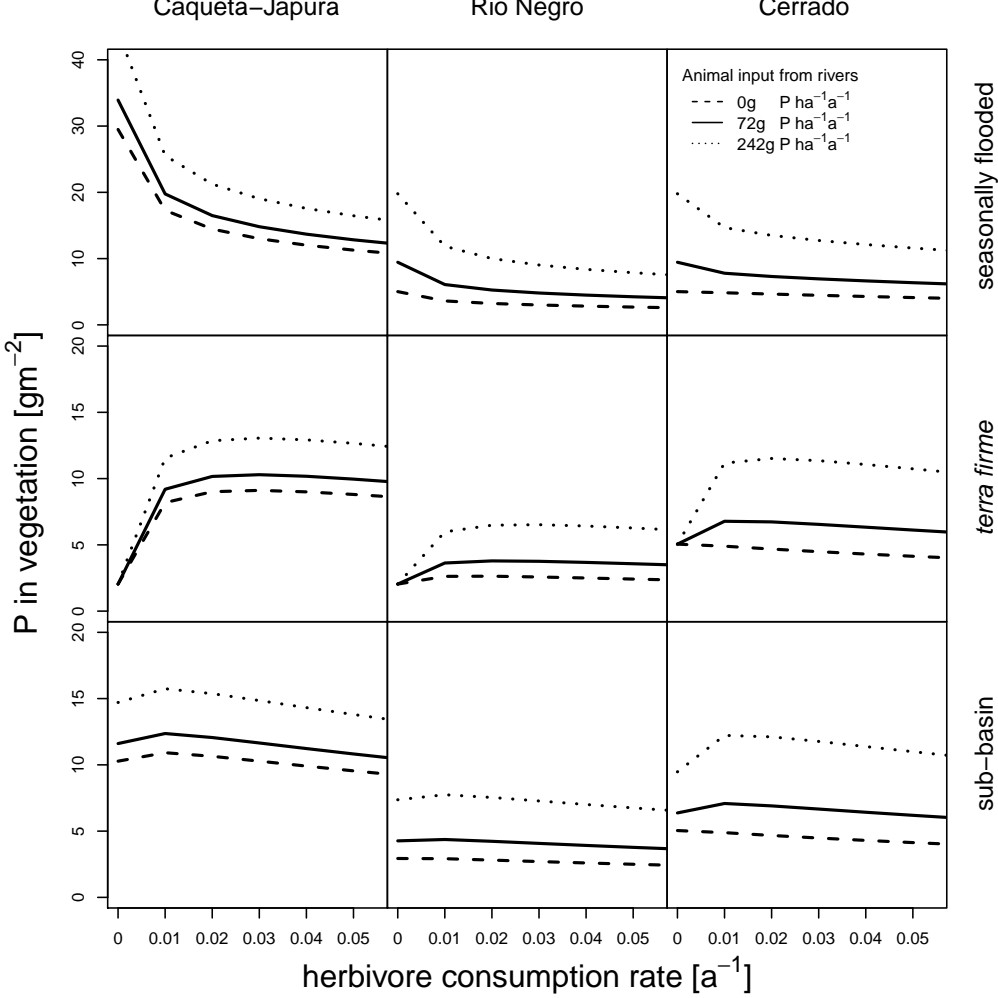

**Figure 4.** Effect of herbivory rates on P in vegetation for the three sub-basins considering flooded and *terra firme* ecosystems. The different line types refer to different input estimates to the flooded ecosystem by terrestrial piscivores (none, otters only, otters and other species). P content at the sub-basin scale is calculated as the area weighted sum for flooded and *terra firme* ecosystems.

and dotted lines compared to the dashed line) significantly improves the P status of both flooded and *terra firme* ecosystems. When testing the combined effect of herbivory and detritivory on the P status of vegetation in the *terra firme* ecosystems, as illustrated in Fig. 6 we dynamics depend on on climate; under humid conditions (Caquetá-Japurá, Rio Negro) both strategies acting simultaneously enhance P in vegetation, while in dry climates (Cerrado) herbivory alone is more effective in enhancing

5  P in vegetation in *terra firme* ecosystems.





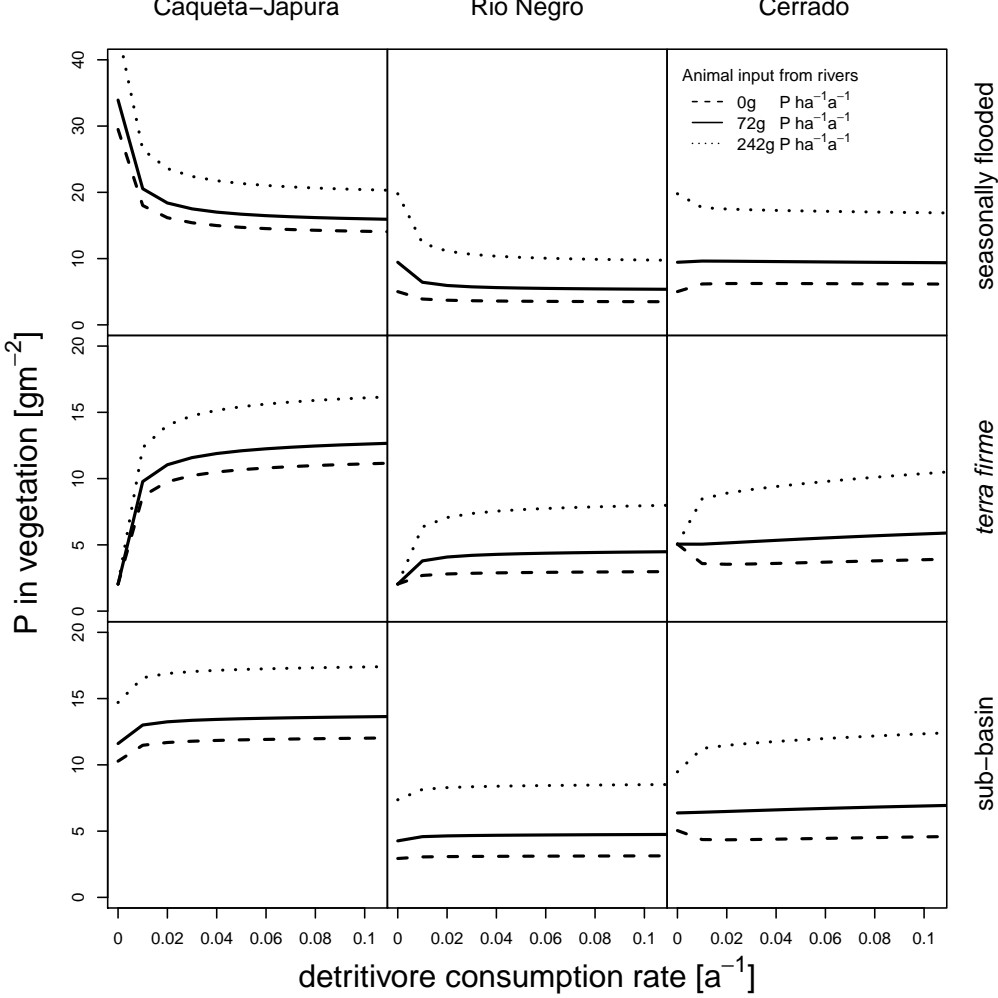

**Figure 5.** Sensitivities of detritivory on P in vegetation for the three sub-basins considering flooded and *terra firme* ecosystems. The different line types refer to different input estimates to the flooded ecosystem by terrestrial piscivores (none, otters only, otters and other species). P content at the sub-basin scale is calculated as the area weighted sum for flooded and *terra firme* ecosystems.

The soil moisture regime has a profound effect on P availability. The Amazon basin includes sub-basins with different environmental settings: some are dry, like Cerrado and others are very humid, like the rain forest. Drier ecosystems have higher amounts of P in vegetation than wet ecosystems with or without herbivory consumption, due to lower P losses (Fig. 7). Soil moisture effects are compounded with redistribution due to herbivory, strongly affecting P in vegetation across ecosystems. In general, less than $1\%$ annual consumption rate by herbivores increases P in vegetation. For dry ecosystems ($s < 0.4$) more than



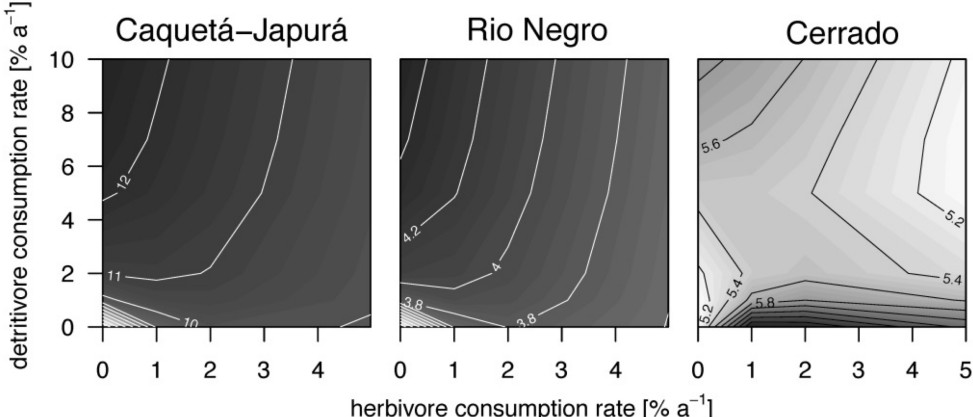

**Figure 6.** Combined effect of herbivory and detritivory on P in vegetation for the three sub-basins (Caquetá-Japurá, Rio Negro, and Cerrado). P in vegetation increases with darkness in the grey scale, the number given in the contour lines represent P in vegetation in $\mathrm{g\,P\,m^{-2}}$) .

1% consumption rate decreases P availability, while for wet ecosystems the maximum is reached at 2% consumption rate by herbivores, which is in agreement with the findings shown in Figures 4 and 5.

## 4   Discussion

We explored the effect of P redistribution through herbivory and detritivory on P availability in three different sub-basins
within the Amazon basin. We also considered different P inputs originating from terrestrial piscivores and we investigated the interaction between herbivory and detritivory, flooding regime, and the role of soil moisture. Our results highlight four important points:

First, plants growing in *terra firme* ecosystems can gain P from redistribution induced by both herbivory and detritivory. Second, animal driven redistribution lead to different results for P in vegetation. While small herbivory rates significantly
enhance the P in vegetation in *terra firme* ecosystems with a maximum around 1-2%, detritivory monotonically increases the P availability in these ecosystems resulting in a saturation at high consumption rates. Third, differences in soil moisture conditions as well as in P input through flooding across the three sub-basins lead to differences in the absolute amount of P in vegetation and to different response to herbivory and detritivory. Fourth, terrestrial piscivores importing P to flooded ecosystems in combination with the redistribution by detritivores and herbivores can fertilize *terra firme* ecosystems. For *terra*
*firme* Cerrados, the extra P input by piscivores switches the redistribution effect of herbivory and detrivory from decreasing to increasing vegetation P.

Our results show that herbivory annual consumption rates o f 1-2 % led to a maximum in P availability in the *terra firme* ecosystem. In a *terra firme* forest in the Rio Negro basin, leaves account for about 0.51 $\mathrm{g\,P\,m^{-2}}$, stems 2.60 $\mathrm{g\,P\,m^{-2}}$, and roots 1.71 $\mathrm{g\,P\,m^{-2}}$ (Uhl and Jordan, 1984), adding up to about 5 $\mathrm{g\,P\,m^{-2}}$. Leaf cutter ants in the tropical forest of Barro Colorado





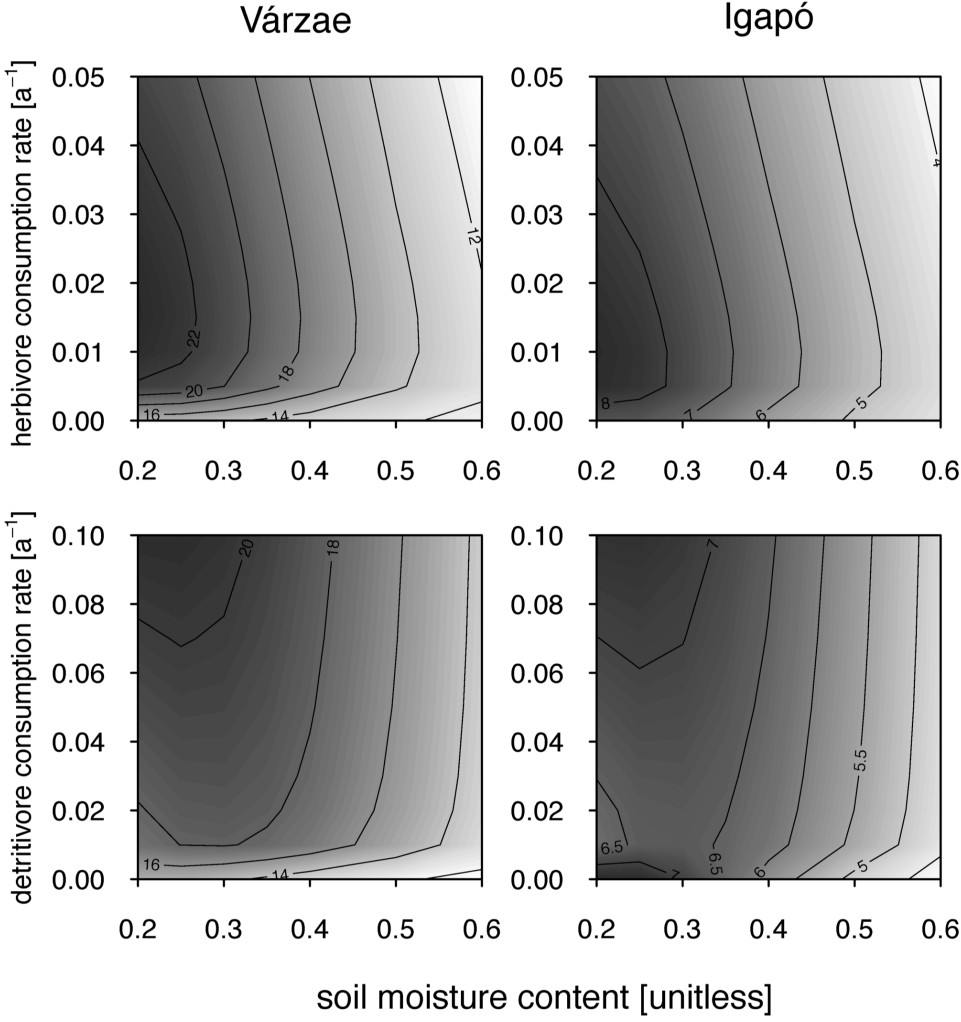

**Figure 7.** The effect of herbivory (upper panels) and detritivory (lower panels) on P in *terra firme* vegetation as dependent on soil moisture and the P in the flooded ecosystems (*Igapó* and *Várzea*). P in vegetation increases with darkness in the grey scale, the number given in the contour lines represents P in vegetation ($\mathrm{g\,P\,m^{-2}}$))

Island (Panama) consume about 10% of foliar biomass per year (Hudson et al., 2009; Metcalfe et al., 2014). Assuming that ant consumption rates are similar across tropical *terra firme* forests, 10% of the foliage consumed in the Rio Negro basin leads to an overall P annual consumption rate of 1% of vegetation biomass. Considering the presence of other herbivores, the overall consumption rate probably ranges between 1 and 3% per year, which is also in agreement with the predicted range that maximizes vegetation P in the *terra firme* ecosystem, and in the upper range of our estimate of 1% herbivory maximizing the P status at the sub-basin scale (Figure 4). Moreover, our model suggests that herbivory rates greater than 2.5% exert a negative effect on P availability in ecosystems that lack substantial sources of P (like the Rio Negro basin). In other words, not in all



cases herbivory is predicted to increase P redistribution. This effect might be important to consider in other models of nutrient redistribution. For example, the role of the mega-herbivores before the megafauna extinction (Doughty et al., 2013). It is worth noting in this context that in about 50% of the world ecosystems, the fraction of biomass consumed by herbivores is indeed lower than 5% (Cebrian and Lartigue, 2004).

Furthermore, our model simulations suggest that P redistribution through detritivores is in general of similar importance than that of herbivores. Observations from central Amazonian forests showing that the proportion of animals feeding on living plant material is rather small (about 7%), while the proportion of animals feeding on detritus is about half of the total (Fittkau and Klinge, 1973) are in agreement with our findings. For example, termites are present in most Amazonian ecosystems (Rückamp et al., 2010), where they abandon nests at a rate of approximately 165 nests $\mathrm{ha^{-1}a^{-1}}$. In terms of P, this rate translates to a

turnover rate of about 600 $\mathrm{g\,P\,ha^{-1}a^{-1}}$, comprising 95% woody turnover and 8.5% total litter turnover (Salick et al., 1983).

These dynamics create micro-sites of fertility, but over larger scales and in the long term offer a mechanism for P transfer from flooded to *terra firme* ecosystems. The study of McKey et al. (2010) on abandoned raised agricultural fields in a seasonal flooded ecosystem found a positive correlation between P content of the fields and the number of termites, ants, and worms. This is consistent with our model finding that redistribution is key to the P budget of the *terra firme* ecosystems and that

detritivories are particularly important in this process (fig. 6). Our results follow from the assumption that herbivores and detritivores can be effectively transport P deep into the *terra firme* ecosystems. Is this assumption reasonable, considering current Amazonian fauna? Amazonian food webs are poorly understood from the perspective of modern science, however, communities inhabiting the Amazon have a deep understanding of them.

Leaf cutting ants, although not herbivores in the strict sense, take amounts of leaves much greater than one would assume

based on their body size. This is because ants do not directly feed on leaves but on the fungi they grow with them. Other animals in turn feed on ants, like birds, anteaters, monkeys, which in turn are eaten by large predators like the jaguar. Thus leaf P can be later excreted by a jaguar in the deepest part of the *terra firme* ecosystems due to the activity of the whole food web. This means that a complex food web may allow P transport to areas far away from seasonally flooded ecosystems and rivers. To illustrate this, one may consider the analogy to a wave: A wave moves over long distances, but the particles (animals in our

context) transfer the energy (P) from particle to particle. It is not necessary that a single animal moves far, but it is the total action of the movement that could result in a net input of P to the P depleted regions. We can imagine how a P atom may travel in complex ways across the Amazon, however it seems impossible to model exactly how it moves.

Animal movement due to dual habitat use is not restricted to invertebrates, but are also documented for fructivorous vertebrates, which use both seasonally flooded and *terra firme* habitats. Some animals move on a daily basis, while others move on

a seasonal basis. Haugaasen and Peres (2007, 2008, 2009) showed that this movement is related to spatial variations in fruit availability. The arboreal species take advantage of the newly available immature and mature fruits, while terrestrial vertebrates mainly profit from fruits remaining after the flood. As we already mention in the introduction, a study on a population of Woolly monkeys (*Lagothrix lagothricha lugens*) shows that the dual habitat use results in a net P flux of 1-4 $\mathrm{g\,P\,ha^{-1}a^{-1}}$ from seasonally flooded forest to *terra firme* forest Stevenson and Guzmán-Caro (2010). The magnitude of P imported by this



single monkey population is comparable to P inputs through the atmosphere originating from the Saharan desert. Hence, the finding of our model that animals contribute substantially to P redistribution in the Amazon basin appears reasonable.

Our results illustrate how terrestrial animals and the associated food webs that feed on riverine sources of food together with herbivores and/or detritivores can fertilize *terra firme* ecosystems. Those P imports are particularly important in sub-basins

drained by clear and black water rivers, which do not receive large amounts of P because the waters that flood those sub-basins are very poor in P. Figure 2 illustrates some of the animals (birds, insects, snakes, and big cats) that are driving the P transfer associated to the Mirití, a black water tributary to the Caquetá, based on the personal experiences of the native artists.

This flow of P associated with the amazonian food webs has implications for human effects on the Amazon basin. Humans have inhabited the Amazon basin for at least 19 000 years and also rely in riverine sources of food. They have created a soil of

high fertility, the *terra preta*, which has been shown to be widespread in the western part of the Amazon and it is associated to clear and black water rivers (McMichael et al., 2014). The population density of pre-Colombian societies in the Amazon before European arrival is still highly uncertain. However, one could imagine that humans are as important as other predators for the transfer of P from rivers to land.

So far, we solely considered the transfer of P from rivers to land, but not the P transfer between sub-basins. Although this is

not included in our model, fish migration in the Amazon River network constitutes an important mechanism transferring P from the nutrient-rich white waters to the nutrient-poor black and clear water rivers (McClain and Naiman, 2008). Large predator species such as the catfish and detrital-feeding fish species migrate from rivers relying on the Andean nutrient supply to rivers that drain nutrient-poor lowlands (McClain and Naiman, 2008; Barthem and Goulding, 1997). Although fish migrations are well studied, the reasons why they occur remains unclear. One reason might be stoichiometric constraints during ontogenesis

(Sterner and Elser, 2002). Juveniles migrate to the estuaries (Barthem and Goulding, 1997) where P is abundantly available due to the mixture of sediments with salt water. There, they can feed on nutrient-rich resources for growth. Adults mainly require energy and locations for reproduction, which are mainly found upstream in small rivers in the forests (Sterner and Elser, 2002). This migration potentially results in a depletion of the P gradient between rivers originating in the Andes, the mouth of the Amazon River, and the black water lowland rivers (McClain and Naiman, 2008). Using information on the productivity of the

forest and how fish changes its nutritional needs though its life cycle could help to better understand this remarkable animal driven P redistribution mechanism.

A publication presenting a model of P redistribution due to herbivory in the Amazon basin by Doughty et al. (2013) argued that the last extinction of mega-herbivores, about 13.000 years ago decreased significantly P redistribution within Amazon basin. They suggest "major human impacts on global biogeochemical cycles stretch back to well before the dawn of agriculture.

Aspects of the Anthropocene may have begun with the Pleistocene megafaunal extinctions". Our results agree with the results of Doughty et al. (2013) in identifying animals as important drivers of the P cycle and therefore essential to Amazon productivity. However, we obtain different insights about P dynamics in the Amazon and how important herbivory and detritivory might be. We find that excessive herbivory can have negative effects of the P budget. Thus the expansion of cattle farming in the Amazon (i.e. human associated mega-fauna) is not only a driver of deforestation but may also have long term effects on

biogeochemical cycling of the amazon. If cattle feeds on vegetation in the basin, but is transported elsewhere for consumption,





it represents a net P loss from the system. Locally, cattle movement can concentrate P around drinking or resting areas, thus substituting the natural redistribution processes with a P concentration mechanism. P would then be easily lost via leaching from these biogeochemical hot spots, which would also represent a net loss, but via a different route. In contrast to our approach, Doughty et al. (2013) solely consider herbivores feeding in the seasonally flooded ecosystems, and base their model on the behavior of current large herbivores living in African savannas. Herbivore movement is approximated by a Brownian motion and the redistribution of P is assumed to be proportional to the size of the herbivore (Doughty et al., 2013; Wolf et al., 2013). Therefore, the effects of detritivores and small organisms, such as leaf cutter ants that harvest a disproportionately large amount of biomass compared to their size, are neglected, and the role of complex food webs that may allow long-distance transport may be underestimated. This model also does not consider terrestrial animals feeding on riverine food sources, like birds, humans, and otters, which our model shows to be very important in terms of P redistribution.

Meanwhile with the extinction of most of the mega-herbivores (as assumed by Doughty et al. 2013) pre-columbian societies would have shifted their diet towards fish, and by this would have enhanced the flux of P from rivers to land. *Terra petra* soils widespread in many *terra firme* ecosystems in western Amazonia are evidence for this human action. Moreover, pre-Colombian societies may have increased the contact areas with rivers by creating ponds and channels (so-called Earthworks, or geoplyphs), which may have increased nutrient input by flooding (Mann, 2008). Taking this one step further, one may speculate whether pre-columbian cultures of the Amazon intentionally enhanced the nutrient flux from river to terrestrial ecosystems and whether they did this by creating channels, feeding primarily on riverine sources of food, and keeping their waste on land.

## 5   Conclusions

We used a simple model to illustrate and discuss our hypothesis that animals may significantly contribute to the internal redistribution of P within the Amazon basin. While rivers tend to dissipate the large scale P gradient between the Andes and the lowlands (McClain and Naiman, 2008), animals do the same across sub-basins and at the landscape scale between river, seasonally-flooded and *terra firme* ecosystems. Our model assumes that the P from the Andes that is redistributed by rivers and animals could prevent Amazon lowland forests from falling into a *retrogressive phase* despite deeply weathered and nutrient poor soils. This is in contrast to previous studies that mainly attribute high Amazonian productivity to exogenous atmospheric P imports. We advocate the view that redistribution processes within the Amazon basin are at least as important as exogenous inputs, based on a synthesis of the available information and a modelling exercise for the three major ecosystem types within Amazonia. Having in mind future empirical tests and investigations, we summarize our results as follows.

Flooding not only provides P, but also takes it away, especially through biomass removal. Therefore not only the strength of the P gradient between seasonally flooded and *terra firme* ecosystem is important but also the soil moisture regime and the duration of flooding when comparing different locations or sub-basins.

Herbivores dissipate the P gradient between seasonally flooded and *terra firme* ecosystems much more efficiently than detritivores, while consumption rates of detritivores can be an order of magnitude higher than consumption rates of herbivores. Herbivory annual consumption rates o f 1-2 % led to a maximum in P availability in the *terra firme* ecosystem. Herbivory and





detritivory are alternative pathways enriching the P content of *terra firme* ecosystems. For understanding P Amazon dynamics it would be important to quantify those effects in the field, e.g. by measuring the consumption rates of herbivores and detritivores along a P gradient.

Taking these insights one step further, landscape changes that are currently occurring in the Amazon region, such as the construction of dams, canalization of the main channels, land transformation towards cattle and agricultural farming and consequent biodiversity losses will have farther impacts on the intermediate and long-term P dynamics of Amazonia and, consequently, its productivity and ecosystems dynamics. For example, the canalization of rivers reduces the contact area between rivers and the terrestrial ecosystems, thereby reducing flood plains that constitute important fertility hotspots. Dams disrupt fish migrations, thus reducing the P flux from nutrient-rich freshwater ecosystems like the Caquetá-Japurá river, to nutrient poor rivers like the Xingu. Fish overexploitation, particularly for export have a similar effect. Land-use change towards cattle farming and monoculture is a major driver of biodiversity loss and may thus reduce the ways P is redistributed across the basin and with that *terra firme* ecosystems productivity.

Therefore, a more detailed exploration of P fluxes associated with animals seems necessary for better understanding the mechanisms that prevent the Amazon region from reaching P depletion. In general, our findings support the view that biodiversity and the proximity to the Andes are essential for the P budget of lowland ecosystems.

*Author contributions.* Statement of authorship: CB, AK, and AP defined the research question, CB and SM designed the model, CB and BR performed model simulations and analyzed the results, CB wrote the first draft of the manuscript, and all authors contributed substantially to revisions.

*Acknowledgements.* C.B. would like to thank Alvaro Buendía, Carlos Rodriguez, Johana Yucuna and Marcela Yucuna for pointing out ways animals transport P, Carlos Sierra for comments on model structure, Lee Miller and Kerry Hinds for comments on manuscript structure and language, Thomas Hickler for comments and corrections of this manuscript, Natalie Mahowald and Carlos Jordan for pointing out some important aspects of the Amazon P cycle, and the Max Planck Society for supporting C.B. with a doctoral scholarship and A.P. by the Agriculture and Food Research Initiative from the USDA National Institute of Food and Agriculture (2011-67003-30222) and the National Science Foundation (CBET-1033467, FESD 1338694, and EAR 1331846 for the Calhoun Critical Zone Observatory).



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
