# Peer review of "Evaluating the effect of nutrient redistribution by animals on the phosphorus cycle of lowland Amazonia"

_Biogeosciences, 2017_

## Referee Comment (RC1) · Anonymous Referee #1 · 16 May 2017

The study aims to assess the importance of redistribution of phosphorus between seasonally flooded and upland (terra-firma) within Amazonian sub-basins by animals (herbivor and detritivor). Different theoretical sub-basins (characterized by different soil water content for upland and different P input from flooding) are studied. The final question is to understand if such proces can contribute to prevent Amazonian ecosystems to fall within terminal steady-state.

The horizontal redistribution of nutrient - P here- by animals is a relevant research question (and, I have to admit, new for me) and I encourage publication. The theoretical framework used to answer to the question is interesting. Both the introduction and discussion are well written. However some clarifications in the Methods section as well as additional description/analysis in the Results section are required. That is why I recommend major revisions before any publication.

**Major comments**

1) I found that the Results section is too short, not totally clear and deserves deeper analysis. More details are given below:
- one of the key process (animal consumption leads to both decrease and increase losses – respectively through redistribution towards terra-firma with lower leaching and towards dissolved P pool prone to higher leaching) quoted p12,L16-17 should be illustrated and more strongly demonstrated: e.g. by using following plots: dissolved/total P ratio as function of the consumption rate, leaching from available P as function of the consumption rate, total leaching of the sub-basin as function of the consumption rate, etc.
- there is no combination humid x poor (see methods and Fig 4 and 5) while it is mentioned on p12,L9 and while continuous values of soil water content for Varzea are used in Fig. 7 left panels. In Methods, only two values are used for soil water content of terra-firma (0.35 and 0.6 given on p11,L14) while a range is given in Table 3.
- the redistribution sensitivity to the transfer from land to river by piscivores is described in two separate paragraphs: p12,L13 and p12,L32. They should be put together.
- p12,L15: the meaning of "optimal" is not clear: do the authors mean *maximum* of biomass in vegetation? for terra-firma or whole sub-basin?
- p12,L19-30: not clear. E.g. the maximum biomass on terra-firma for rate of 0.2% mentioned L19 applies only to Caqueta-Japura? What explains the "difference of 1%" (L20) is not clear. The role of gradient between flooded and terra-firma and the role of the leaching is mentioned but not demonstrated (see above). The authors should refer to the Fig.7 in this section (this figure underlines the role played by soil water content). " In contrast" (L25) does not make sense because previous sentence focuses on whole sub-basin while the following sentence focuses on terra-firma. Why "redistribution causes more losses than gains" (L26) ? What explains one major finding (for dry x poor combination, terra firma has larger P than flooded area) is not clear.
- Fig.4: what is the default value used for detritivore consumption rate? Fig.5: what is the default value used for herbivore consumption rate? This has to be given in the Method section.
- Fig.4 and 5: the authors should show on the same plot the P in vegetation for seasonally flooded and terra-firma. This will show more clearly that terra-firma > flooded area on Cerrado sub-basin.
- Figs.4 and 5: remind "dry", "humid", "rich", "poor" on the different line/column titles to help the reader
- Fig.6: the fact that fig 6 focuses on P in vegetation of terra-firma (as explained on p13,L2) is missing in the caption.
- the interpretation/reading of some Figures are not straightforward: E.g.: p13,L4: "in dry climates (Cerrado) herbivory alone is more effective in enhancing P in vegetation in terra firme ecosystems.". I cannot read these results from Fig.6: for a given detritivore consumption rate, increasing the herbivore consumption rate (go from left to right on a horizontal line) does not increase P in vegetation (even slight decrease).

2) Error or lack of clarity in the equation describing the redistribution of total herbivor/detritivor consumptions between ecosystems (flooded and terra-firma)

p6,L26: " (…) herbivores consume (...), whereas detritivores consume (…). These fluxes are then returned to the available P and detritus P compartments in the seasonally flooded (F) and terra-firme ecosystems (U). Each ecosystem receives a fraction of the total consumption equal to its fractional area ($A_F$ and $A_U$, respectively)."
I totally agree with this sentence: the fraction of total (from both flooded and terra-firma) herbivore consumption that returns to a given ecosystem (flooded or terra-firma) has to be equal to the ratio between this ecosystem area and the total sub-basin area (either $A_F$ or $A_U$). However, this does not appear in the equations given in the Method section.
Basically, $AI_{vdE}$ should be equal to $A_E.k_{HM}.k_H.(A_U.P_{vU} + A_F.P_{vF})$ (and not $k_{HM}.k_H.(A_U.P_{vU} + A_F.P_{vF})$). The term $k_{HM}.k_H.(A_U.P_{vU} + A_F.P_{vF})$ should be equal to $AI_{vdU}+AI_{vdF}$.

This perhaps arises from a confusion between equations given in the text [some of them, such as $k_H.(A_U.P_{vU} + A_F.P_{vF})$, are multiplied by an area (unitless)] and fluxes described in Fig.3 and Table 1 (e.g. $AO_{vU}=k_H.P_{vU}$). Overall, I found the section p6,L20-30 difficult to follow because of the huge numbers of variables introduced, which are not totally consistent with Fig.3.

3) For a given sub-basin, the authors restrict their sensitivity analysis to the animal consumption rates and animal input from rivers (Fig.4 and 5) while it would be worth assessing the potential role played by other variables. The theoretical framework built by the authors is particularly appropriate to this. In particular, it would be interesting to study the P redistribution sensitivity to:
- the fraction of the sub-basin covered by each ecosystem ($A_U$, $A_F$) (fixed values of 70 and 30% in the current study)
- a difference in $k_H$ between flooded and terra-firma (e.g. to describe difference in population densities or in vegetation biomass between the two ecosystems)
- a difference in soil properties between the two ecosystems that could modulate occlusion or leaching rates. It is true that difference in leaching rate is already taken into account through its sensitivity to the soil water content but what the effect of a difference in soil properties could be?
- the magnitude of $I_f$ (for a given sub-basin) to described some variation in flood pulses, flood duration, etc. E.g. some plot showing the redistribution efficiency vs. $I_f$ would be interesting.

4) Many parameters are uncertain (see section 2.2 and Table 2) and it would be particularly interesting to understand how this uncertainty propagates to the final P redistribution within the sub-basin. In particular, could some processes considered as negligible right now be underestimated?

5) The abstract does not reflect properly the findings of the study:
- please, remove "between sub-basins" (L5) and "fish migrations" (L9) because they do not correspond to the focus of this study (e.g. p18,L14: "Although this is not included in our model, fish migration")
- develop the key-results (e.g. summarize findings given p15,L8-16)

6) The final question is to understand if animals P re-distribution can contribute to prevent Amazonian ecosystems to fall within terminal steady-state. This is mentioned in the discussion (p19,L23). However, more analysis is required in the Results section: e.g. do the simulations without animal redistribution (the ones at the left-bottom corner of Fig.6) reach this terminal steady-state?

**Minor comments and type-setting**

- p15,L11: meaning of "saturation"?
- the authors should remind to the reader how the leaching is computed given its role to explain the difference between dry and humid sub-basins
- the authors should justify that the Results focus on P in vegetation (because of the final question about the terminal steady-state?)
- the redistribution of nutrient has also been studied between cropland and forest in temperate ecosystems (see e.g. (Abbas *et al.*, 2012)) and could be quoted in the discussion?

p1,L9: " interweaved" cannot be understood at that stage (but only after reading p17)
p3,L2: " how they could be reaching" → "how they could reach"?
Fig.1: meaning of dashed vs. solid arrows?
p3,L5: not clear how this "contradicts"?
p3,L27: "Overall, for the terrestrial ecosystems of the Amazon basin the atmosphere could even act as P sink, rather than a net P source.": not clear from what is explained before in the paragraph.
p4,L15: "typically against the gradients of physical flow processes.": not clear
p4,L22: the status of the piscivory is not clear in the introduction: "piscivory" is quoted besides "herbivory" and "detritivory" then not mentioned at all in the introduction.
p6,L20: "As consider in this model" → "Two … processes are considered in this model:"
p6,L23: food web → food webs
p6,L22: say explicitly here that the subscript E corresponds to either F or U. It appears only on p8 (also in caption of Table1)
p9,L2: "and occlusion $F_{dCE}$"? Occlusion corresponds to $O_{oU}$ in Fig.3.
Equation 3: subscript E is missing in $P_d$, $I_d$, $I_w$, etc.
p10,L18: missing "as" before "the Rio Negro sub-basin"
p11,L12: "We run the model for terra firme ecosystems (U) using yearly averaged relative soil water content of 0.35 for the Cerrado (Runyan and D'Odorico, 2012), and 0.6 for the terra firme part of the Caquetá-Japurá and Rio Negro sub-basins." → "We run the model for terra firme ecosystems (U) using yearly averaged relative soil water content of 0.35 for the Cerrado (Runyan and D'Odorico, 2012), and 0.6 for the Caquetá-Japurá and Rio Negro sub-basins."
p11,L31: not clear how the upper limit for input from river to lands by animals (242gP/ha/a) is chosen.
Table3: "variable to 1957", "variable to 250": does it mean that the value given in the column "value" is the lowest boundary? Or default value?
p12,L6: "(4 vs. 5)" → "(Fig.4 vs. Fig.5)"
Fig.4 and 5: remind the name of the variables used in the plots ($k_H$, $k_D$, $I_{aF}$)
p12,L9: "than their associated to várzea ecosystems" → "than terra-firma associated to varza ecosystems"?
p12,L13: the authors should mention "piscivores"
P13,L3: "we"?; "on on"
Table 1: missing "of" in column "description" for line "animal fluxes"

Ref:
Abbas F, Merlet J, Morellet N *et al.* (2012) Roe deer may markedly alter forest nitrogen and phosphorus budgets across Europe. *Oikos*, **121**, 1271–1278.

---

## Referee Comment (RC2) · Anonymous Referee #2 · 28 May 2017

This manuscript attempts to show the importance of spatial redistribution of P from rivers to land, among sub-basins to maintain P cycles of lowland Amazonia. This is an interesting challenge to understand the P cycles from the view of functional roles of animals. Therefore, this manuscript would attract many readers' attention. The introduction section is generally well written and explains the characteristics of Amazonian ecosystems and the P cycles. However, as described below, there are several concerns to be addressed before recommendation can be made for publication in Biogeosciences.

Main comments: In Introduction, the authors emphasize the dynamic nature of nutrient cycles in Amazon, particularly focusing on the lateral P transfer between different

ecosystems or different sub-basins by terrestrial animals. For example, the authors mention that this study shows animals (herbivores and detritivores) redistribute P from flooded sub-basins to P-poor terra firme sub-basins (e.g., P4L20). However, it is unclear at least to me whether this manuscript actually explores the lateral P transfer. It seems that the authors examine how herbivory and detritivory, and the resultant P mineralization affect the amount of vegetation P within "each" sub-basin of three ecosystems that differ in P availability and precipitation (Fig.4 and 5). The mathematical model of this study seem to include only terrestrial piscivores as a vector causing lateral transport of P from river to land, and the results look very confirmatory. If my understanding is correct, I would like to recommend the authors to rewrite and reorganize the manuscript largely to clarify the rationale and the goal of this study.

In addition, it would be necessary to reorganize Discussion section because most of the sentences seem not relevant to the findings of this study. Please discuss the present results by referring to earlier studies on herbivory and detritivory rates as well as ecosystem properties associated with P cycles in Amazon and other ecosystems. To do this, some sentences could be brought from Introduction section.

Minor comments:

Materials and Methods

P8L8(Fig.3): If P transport from flooded area to non-flooded area (terra firme) is not modeled in this study, it would be better to depict the two boxes representing herbivores or detritivores separately for each area.

P8L15: Please explain the equation and each symbol more carefully here. What OoE stands for?

P10L3: discussed

P10L10-16: These sentences have been described already in Introduction, and thus could be deleted.

P10L17: It would be needed to explain about Rio Negro basin and its relation to the overall Amazon here.

P10L21: I could not understand how the value 1895.84 molP per km2 per year was calculated. Please explain.

P11L3: A reference should be needed here.

P11L4-11: These sentences seem to have been explained already, and thus redundant.

Results

P12L6: What 4 vs.5 stands for?

P12L16: From where is P transported to terra firme ecosystem? If it comes from terra firme vegetation itself, the expression "transport" should be misleading.

P13Figure.4: Please change the order of the panels so as to match the order of site explanations in text (1, Rio Negro, 2, Caqueta-Japura, 3 Cerrado). And, add (%) to the x axis label as in Figure 6.

P13L3: delete "we"

P14L5: What does "s<0.4" mean?

Discussion

P15L8: Starting new paragraph here should not be necessary.

P15L9: different results between what?

P17L2: redistribution, for example, . . ..

P17L16: Again, it is unclear from where the animals transport P into terra firme ecosystems. Please explain.

P17L17-18:This sentence could be deleted.

P17L19-P18L6: These sentences could be deleted, because they are not directly relevant to the present results, and seem very speculative.

P18L6-7: The sentence seems lack of scientific basis.

P18L14-P19L17: These sentences concerning human impacts and megafauna should not be the main topics of this manuscript, and therefore could be deleted.

---

## Author Comment (AC1) · 14 Oct 2017

Referee # 1

The study aims to assess the importance of redistribution of phosphorus between seasonally flooded and upland (terra-firma) within Amazonian sub-basins by animals (herbivor and detritivor). Different theoretical sub-basins (characterized by different soil water content for upland and different P input from flooding) are studied. The final question is to understand if such processes can contribute to prevent Amazonian ecosystems to fall within terminal steady-state. The horizontal redistribution of nutrient - P here- by animals is a relevant research question (and, I have to admit, new for me) and I encourage publication. The theoretical framework used to answer the question is interesting. Both the introduction and discussion are well written. However some clarifications in the Methods section as well as additional description/analysis in the Results section are required. That is why I recommend major revisions before any publication.

We thank you for the great effort put into critically reviewing our manuscript, taking the time to revise in detail our model equations and check for inconsistencies.In our revisions, we put effort into making our rationale and model analyses more clearer, so that they can be easily followed. We modified the explanation of the P redistribution equations, and show that the original equations were correct. The additional analyses suggested are included in the new version of the paper.

Major comments 1) I found that the Results section is too short, not totally clear and deserves deeper analysis.

We agree, therefore, taking into account your comments we rewrote the results section, starting with the plots suggested and some additional ones, explaining each figure in detail and clarifying the issue with the enhancement of P losses due to herbivory and detritivory. The results section was almost entirely modified and therefore changes are not highlighted.

More details are given below: - one of the key process (animal consumption leads to both decrease and increase losses – respectively through redistribution towards terra-firma with lower leaching and towards dissolved P pool prone to higher leaching) quoted p12,L16-17 should be illustrated and more strongly demonstrated: e.g. by using following plots: dissolved/total P ratio as function of the consumption rate, leaching from available P as function of the consumption rate , total leaching of the sub-basin as function of the consumption rate, etc.

In Figures 4 and 5 we present the state variables and fluxes in relation to herbivory and detritivory. The first row of panels in Figure 4 shows averaged P concentration in vegetation per square metre of terra firme ecosystems. The second row shows P

gradients between the seasonally flooded and the terra firme ecosystem, that is A_F x P_vF - A_U P_vU. The last row shows the averaged total P in the sub-basin in gP /m^2, which is estimated: A_F ( P_vF + P_oF + P_dF ) + A_U (P_vU + P_oU + P_dU). Figure 5 shows other relevant fluxes, the first row the net P transfer by animals (herbivory + detritivory) to the terra firme ecosystem (total animal input to TF - animal output of TF). The second row shows the P dissolved losses fromthe terra firme ecosystem (g P /m2 /a). The last row shows the total losses of P at the subbasin scale, also given in g P /m2 /a.

- there is no combination humid x poor (see methods and Fig 4 and 5) while it is mentioned on p12,L9 and while continuous values of soil water content for Varzea are used in Fig. 7 left panels. In Methods, only two values are used for soil water content of terra-firma (0.35 and 0.6 given on p11,L14) while a range is given in Table 3.

The dynamics modelled for Rio Negro subbasin are humid and the rivers carry waters poor in P, thus this case is considered. In the methods section in Table 3 we provide the parametrization of each of the sub-basins. Because the number of figures increased and Figure 7 showing the sensitivity to soil moisture is not crucial and it was difficult for the reader to understand it we decided to remove the figure from the paper. Because Fig. 4 and 5 are new to the revised version of the manuscript improving the results section we decided to remove the Figure about soil moisture which we consider not crucial for the main objective of the manuscript, which is about the P status of vegetation.

- the redistribution sensitivity to the transfer from land to river by piscivores is described in two separate paragraphs: p12,L13 and p12,L32. They should be put together.

In the new result section it only appears once.

- p12,L15: the meaning of "optimal" is not clear: do the authors mean Maximum of biomass in vegetation? for terra-firma or whole sub-basin?

Yes, this definitely requires a definition therefore the explanation in the result section

was modified, special attention was placed to solve this issue. Figure 4 and 5 also show the the P states and fluxes of the different sub-basins and the P status of the flooded ecosystem, so that the reader can better check the fluxes and overall dynamics. We have also added variable names to the figures to help matching equations and result presentation.

- p12,L19-30: not clear. E.g. the maximum biomass on terra-firma for rate of 0.2% mentioned L19 applies only to Caqueta-Japura? What explains the "difference of 1%" (L20) is not clear. The role of gradient between flooded and terra-firma and the role of the leaching is mentioned but not demonstrated (see above).

The difference between the maximum given at the sub basin scale and at terra firme ecosystems occurs as the sub-scale values are calculated as a weighted average of terra firme and seasonally flooded. In the new version of the MS this is explained as follows:

While a rate of herbivore consumption of 1-2% maximizes P in living biomass of the terra firme ecosystem, a rate of only 1% or less maximizes the P status of the whole sub-basin (Figure 5, bottom row), and this maximum only occurs at low detritivore consumption rates. This difference originates from the P gains of the terra firme ecosystem, but also takes into account the losses of the seasonally flooded ecosystem that occupies 30% of the sub-basins area. Therefore, despite larger vegetation P under certain combinations of consumption rates, the steady state total P stocks at the sub-basin scale tend to decrease with increasing consumption rates, except at very low consumptions rates (Figure 5, bottom row), due to the corresponding increasing P losses.

The authors should refer to the Fig.7 in this section (this figure underlines the role played by soil water content). " In contrast" (L25) does not make sense because previous sentence focuses on whole sub-basin while the following sentence focuses on terra-firma. Why "redistribution causes more losses than gains" (L26) ? What explains

one major finding (for dry x poor combination, terra firma has larger P than flooded area) is not clear.

"In contrast" was referring to the dynamics of a wet sub-basin with a dry sub-basin, however, the whole paragraph disappeared as we have rewritten the section to explain the findings in a more clear way. Please refer to the new result section.

- Fig.4: what is the default value used for detritivore consumption rate? Fig.5: what is the default value used for herbivore consumption rate? This has to be given in the Method section.

For Fig. 4 no detritivory is considered, and for Fig. 5 no herbivory is considered. We have included this information in the captions of these figures. Now the figures are numbered as 6 and 7.

- Fig.4 and 5: the authors should show on the same plot the P in vegetation for seasonally flooded and terra-firma. This will show more clearly that terra-firma > flooded area on Cerrado sub-basin.

This information is provided in the first two rows of the figure, we hope that with the new figure caption this point is resolved. The figure numbers are now 6 and 7.

- Figs.4 and 5: remind "dry", "humid", "rich", "poor" on the different line/column titles to help the reader

Sub-basins names and variable names are now included in the figures names.

- Fig.6: the fact that fig 6 focuses on P in vegetation of terra-firma (as explained on p13,L2) is missing in the caption. - the interpretation/reading of some Figures are not straightforward: E.g.: p13,L4: "in dry climates (Cerrado) herbivory alone is more effective in enhancing P in vegetation in terra firme ecosystems.". I cannot read these results from Fig.6: for a given detritivore consumption rate, increasing the herbivore consumption rate (go from left to right on a horizontal line) does not increase P in vegetation (even slight decrease).

Our figure 6 is now included in figure 5 as the top row and with this we put the figure into context, figure 4 is showing some of the fluxes that we consider important so that the reader can have a better understanding on how the model and the processes are coupled. The variable name P_vU is on the right side of the plot.

2) Error or lack of clarity in the equation describing the redistribution of total herbivor/detritivor consumptions between ecosystems (flooded and terra-firma) p6,L26: " (...) herbivores consume (...), whereas detritivores consume (...). These fluxes are then returned to the available P and detritus P compartments in the seasonally flooded (F) and terra-firme ecosystems (U). Each ecosystem receives a fraction of the total consumption equal to its fractional area (AF and AU, respectively)."

I totally agree with this sentence: the fraction of total (from both flooded and terra-firma) herbivore consumption that returns to a given ecosystem (flooded or terra-firma) has to be equal to the ratio between this ecosystem area and the total sub-basin area (either AF or AU). However, this does not appear in the equations given in the Method section. Basically, AIvdE should be... http://www.biogeosciences-discuss.net/bg-2017-121/bg-2017-121-RC1-supplement.pdf

Overall, I found the section p6,L20-30 difficult to follow because of the huge numbers of variables introduced, which are not totally consistent with Fig.3.

We understand this part might be confusing, and tried to unify terms to make it easier to follow. Considering that the values that are calculated are all already in g P /m2/a it would not be appropriate to multiply the flux by AIvdE. For the calculation of the fluxes and states of the sub-basin we apply weighted averaging to take into consideration the different areal extent of terra firme and flooded ecosystems. We have added some sentences in the description of the model to clarify that point and they are highlighted in green.

3) For a given sub-basin, the authors restrict their sensitivity analysis to the animal consumption rates and animal input from rivers (Fig.4 and 5) while it would be worth

assessing the potential role played by other variables. The theoretical framework built by the authors is particularly appropriate to this. In particular, it would be interesting to study the P redistribution sensitivity to:

- the fraction of the sub-basin covered by each ecosystem (AU, AF) (fixed values of 70 and 30% in the current study) - a difference in kH between flooded and terra-firma (e.g. to describe difference in population densities or in vegetation biomass between the two ecosystems) - a difference in soil properties between the two ecosystems that could modulate occlusion or leaching rates. It is true that difference in leaching rate is already taken into account through its sensitivity to the soil water content but what the effect of a difference in soil properties could be? - the magnitude of If (for a given sub-basin) to described some variation in flood pulses, flood duration, etc. E.g. some plot showing the redistribution efficiency vs.If would be interesting. Regarding this point: - the fraction of the sub-basin covered by each ecosystem (AU, AF) (fixed values of 70 and 30% in the current study)

In the methods sections we added some lines: Interestingly, the more uniform the partition between flooded and upland ecosystems, the larger the flux, because AU $(1 - AU)$ is maximized at AU = AU = 0.5. The same reasoning can be applied to detritivory, and the corresponding equations are obtained by substituting the subscript H by D.

Additional, sensitivities to soil moisture, uplift rates, active uptake, losses, occlusion, were also shown in an earlier version of the model (Buendía et al, 2010 https://www.biogeosciences.net/7/2025/2010/bg-7-2025-2010.pdf ); additional work was put in the paper Buendía et al., 2014, in which we have implemented the 2010 model to a model running on the global scale couple to a soil formation model to test the importance of deocclusion, active uptake and relate the model to carbon dynamics. There we tested the potential of de-occlusion in the longterm to serve as P source.

We are interested in testing more sensitivities and in the process of adjusting and

developing this version of the model we have done some of the simulations you are suggesting, however, as indicated by the comments of both reviewers, the manuscript is already structurally complex and we would prefer to keep it streamlined to leave some space for discussions on result implications. In that sense choices have to be made regarding what to show; here we have focused on the role of herbivory, detritivory and the implications of piscivores inputs on the P status of terra firme vegetation.

4) Many parameters are uncertain (see section 2.2 and Table 2) and it would be particularly interesting to understand how this uncertainty propagates to the final P redistribution within the sub-basin. In particular, could some processes considered as negligible right now be underestimated?

Some lines on Af and AU are include in the new versión of the model and in Buendía et al. 2010 sensitivity analises to most the other parameters were performed.

5) The abstract does not reflect properly the findings of the study: - please, remove "between sub-basins" (L5) and "fish migrations" (L9) because they do not correspond to the focus of this study (e.g. p18,L14: "Although this is not included in our model, fish migration") - develop the key-results (e.g. summarize findings given p15,L8-16)

We have reorganized the paper and modified the abstract to make it clear that although our model does not deal with migration, our discussion and synthesis of the main processes occurring in the Amazon supports the idea that this process has important implications. We have removed "migration" from the abstract, as it was rewritten also to take into account our second reviewer comments.

6) The final question is to understand if animals P re-distribution can contribute to prevent Amazonian ecosystems to fall within terminal steady-state. This is mentioned in the discussion (p19,L23). However, more analysis is required in the Results section: e.g. do the simulations without animal redistribution (the ones at the left-bottom corner of Fig.6) reach this terminal steady-state?

In Buendía et al. (2010) we claimed that tectonic uplift and weathering were important processes for the Amazon basin, therefore we could not see the terminal steady-state as it might occur in Hawaii where the oceanic crust is heavier than the continental crust and therefore,once the hotspot is not supporting further growth of the island, there are no other forces refreshing parent material for further weathering. In this paper, sensitivities analyses to dust deposition were performed 0.0011–0.0048 g P /m2 /a (Swap et al., 1992) similar to what is assumed in this version of model. While weathering was estimated by the model using a concentration of 100 mol P/m3 in parental rocks, which in steady state solution leads to a weathering release of P of about 110 g P /m2 /ha, 10 g P v/m2, higher than the value we used in our current model. The current model and simulations now used the steady state solution, however parameterized according to Gardner et al. (1990) , which is about 80 g P /m2 /ha. This P concentration in the parent material is a parameter that does not change the general dynamics, but it reduces the amount of P in vegetation, and with that, the possibility to support vegetation growth. Ecosystems have a very flexible C to P ratio and therefore it is hard to set the line were limitation starts as it depends on the ecosystems composition and other factors. A study by Jordan et al. (XX) in the rionegro sub-basin estimated a P concentration in vegetation of 5 g P /m2. Our model without animal transport predicts 2.5 g P/ m2, in vegetation in the terra firme, but with about 200 g P /ha/a and herbivory redistribution of more than 1% or riverine input of about 72 and 8 % detritivory 5 g PvU/m2 are achieved. A study from the Cerrados reporting on a 26 year fire exclusion reported 2.16 g P / m2 in vegetation, and 25.9 g P /m2 in organic material (Resende et al., 2011). One important factor is fire, due to the expansion of the agricultural frontier in Amazonia, which produces ashes that are redistributed across the basin and are fertilizing "P-depleted places", suggesting that atmospheric observations and measurements taken today should consider this factor

Minor comments and type-setting

- p15,L11: meaning of "saturation"? - the authors should remind to the reader how the
leaching is computed given its role to explain the difference between dry and humid sub-basins

Leaching is computed and calibrated in Buendía et al., 2010. We hope that withe the new result section this becomes more clear.

- the authors should justify that the Results focus on P in vegetation (because of the final question about the terminal steady-state?)

We hope that with the new structure of the paper that becomes clear.

- the redistribution of nutrient has also been studied between cropland and forest in temperate ecosystems (see e.g (Abbaset al., 2012)) and could be quoted in the discussion

We did not want to add more things to the already complex paper

p1,L9: " interweaved" cannot be understood at that stage (but only after reading p17)

The abstract was rewritten, the new version does not include the word interweaved

p3,L2: " how they could be reaching" → "how they could reach"?

Changed

Fig.1: meaning of dashed vs. solid arrows?

The dashed arrows represent the animal driven processes. Now this is included in the main text and we also explicitly refer to herbivory and detritivory in Fig. 1

p3,L5: not clear how this "contradicts"?

The text was modified to:

Weathering in the central Amazon basin was estimated to be about 75 g P ha−1a−1 based on data taken at the mouth of Rio Negro river, which is an important tributary of the Amazon river draining only the lowlands (Gardner, 1990). This measurement contrast to trends observed in soil chronosecuences like Hawaii Islands and Franz Joseph glacier retrogression (Walker and Syers, 1976; Wardle, 2004; Wardle et al., 2009), where at terminal steady state no weathering is detectable. The Amazon basin experiences continental isostatic rebound, where the slow erosion rates are compensated by slow uplift and weathering of new material (Porder et al., 2007; BuendiÌĄa et al., 2010). However, because bedrock can be as deep as 100 m, it is not clear whether or not P released by weathering can reach the terrestrial biotic cycle (Gardner, 1990). Nevertheless, as dissolved P reaches the rivers by ground-water flow, it can be used by freshwater ecosystems and redistributed across sub basins with floods.

p3,L27: "Overall, for the terrestrial ecosystems of the Amazon basin the atmosphere could even act as P sink, rather than a net P source.": not clear from what is explained before in the paragraph.

Biogenic particles, such as pollen, spores, bacteria, algae, protozoa, fungi, and leaf fragments are generated by the forest and although to a great extent most of them are deposited in the forest again, some (about 19 \,\unit{g\,P\,haˆ{-1} aˆ{-1}}) fall into the Altlantic ocean\citep{Mahowald:2005}, where they become an important nutrient source to Atlantic marine ecosystems near the continent. Fires caused by the amplification of the agricultural frontier also contribute to the internal P redistribution and export \citep{Artaxo:1994,Mahowald:2005,Pauliquevis:2012gx}. Therefore, according to some of these estimates, the atmosphere could even drive more P losses than inputs to the amazon basin.

p4,L15: "typically against the gradients of physical flow processes.": not clear

Changed to: moving P even against the topographical gradient which however drives water and matter flows lowlands.

p4,L22: the status of the piscivory is not clear in the introduction: "piscivory" is quoted besides "herbivory" and "detritivory" then not mentioned at all in the introduction.
A rough estimate on piscivory (otter P imports to land even estimated) is given in the introduction. Fig. 1 was modified to explicitly refer piscivores and now the is a sub-subsection referring to it in the model description.

p6,L20: "As consider in this model" → "Two ... processes are considered in this model:"

Changed

p6,L23: food web → food webs

Changed

p6,L22: say explicitly here that the subscript E corresponds to either F or U. It appears only on p8 (also in caption of Table1)

Now this clarification is introduced at the beginning of the model description.

p9,L2: "and occlusion FdCE"? Occlusion corresponds to OoUin Fig.3. Equation 3: subscript E is missing in Pd, Id, Iw, etc.

Thanks for the observation. Id and Iw have the same value, but we modified the other subscripts and correct the inconsistencies.

p10,L18: missing "as" before "the Rio Negro sub-basin"

Changed

p11,L12: "We run the model for terra firme ecosystems (U) using yearly averaged relative soilwater content of 0.35 for the Cerrado (Runyan and D'Odorico, 2012), and 0.6 for the terra firme part of the Caquetá-Japurá and Rio Negro sub-basins." → "We run the model for terra firme ecosystems (U) using yearly averaged relative soil water content of 0.35 for the Cerrado (Runyan and D'Odorico, 2012), and 0.6 for the Caquetá-Japurá and Rio Negro sub-basins."

changed.

p11,L31: not clear how the upper limit for input from river to lands by animals

(242gP/ha/a) is chosen.

The text was modified to:

The first value simulates a scenario with no animals, the second simulates a scenario in which P transfer is like the one estimated for giant otter (Pteronura brasiliensis; see calculation in the introduction), and the last one simulates a scenario in which otters and other animals contribute; since this contribution is unknown, the limit for the sensitivity analysis was set to a value between 3 to 4 times the second estimate, 242 g P ha−1 a−1 .

Table3: "variable to 1957", "variable to 250": does it mean that the value given in the column"value" is the lowest boundary? Or default value?

More explanation is provided in the text:

Considering that most of the material is transported during the raining season, that plains are innundaded some months of the year and P can recycle within the basin more times before it is discharged into the ocean, we let the flooding input for the vaÌA̧rzea Ifw be 80% of the estimated, 1566 gP ha−1a−1 and the flooding input to the igapoÌA̧ ffB 90% of estimated 196 g P ha−1a−1.

p12,L6: "(4 vs. 5)" → "(Fig.4 vs. Fig.5)" Fig.4 and 5: remind the name of the variables used in the plots (kH, kD, IaF)

It was changed

p12,L9: "than their associated to várzea ecosystems" → "than terra-firma associated to varza ecosystems"?

Changed to: than their associated varza ecosystems.

p12,L13: the authors should mention "piscivores"

Piscivores are now widely explained throughout the text.

P13,L3: "we"?; "on on" Changed

Table 1: missing "of" in column "description" for line "animal fluxes Included

Please also note the supplement to this comment:
https://www.biogeosciences-discuss.net/bg-2017-121/bg-2017-121-AC1-
supplement.pdf

---

## Author Comment (AC2) · 15 Oct 2017

Reviewer # 2 This manuscript attempts to show the importance of spatial redistribution of P from rivers to land, among sub-basins to maintain P cycles of lowland Amazonia. This is an interesting challenge to understand the P cycles from the view of functional roles of animals. Therefore, this manuscript would attract many readers' attention. The introduction section is generally well written and explains the characteristics of Amazonian ecosystems and the P cycles. However, as described below, there are several concerns to be addressed before recommendation can be made for publication in BioGeosciences.

[Figure]

We thank you for reviewing our manuscript and for recommending re-structuring the paper. We hope that by substantially re-writing large sections we addressed your concerns.

Main comments: In Introduction, the authors emphasize the dynamic nature of nutrient cycles in Amazon, particularly focusing on the lateral P transfer between different ecosystems or different sub-basins by terrestrial animals. For example, the authors mention that this study shows animals (herbivores and detritivores) redistribute P from flooded sub-basins to P-poor terra firme sub-basins (e.g., P4L20). However, it is unclear at least to me whether this manuscript actually explores the lateral P transfer. It seems that the authors examine how herbivory and detritivory, and the resultant P mineralization affect the amount of vegetation P within "each" sub-basin of three ecosystems that differ in P availability and precipitation (Fig.4 and 5). The mathematical model of this study seem to include only terrestrial piscivores as a vector causing lateral transport of P from river to land, and the results look very confirmatory. If my understanding is correct, I would like to recommend the authors to rewrite and reorganize the manuscript largely to clarify the rationale and the goal of this study.

This is a central point in our manuscript - indeed our aim with this simple model is to describe in a lumped way lateral P transfer within sub-basins. Two modes of transport are considered - both connecting lowlands and uplands and therefore representing 'lateral' transport. The first mode is by non-directional redistribution of P by herbivores and detritivores that respectively feed on vegetation and litter P in proportion to the amounts of P in those pools. By returning P to either lowlands or uplands in proportion to the areas of these ecosystems, they effectively redistribute P across the landscape, though not in a pre-specified direction. The direction of P flow, in fact, is emerging from the model-generated gradients in P across the landscape. The second mode of transport is the one mentioned by the reviewer: piscivores feeding in the rivers and excreting (or dying) on land. We assessed the role of both modes of P transfer in Figures 4-7. The new Figures 4-5 should provide a better illustration of the first P

transfer mode compared to the previous version of the manuscript. The text has also been re-structured to clarify which mechanisms of P transport are considered in the model.

In addition, it would be necessary to reorganize Discussion section because most of the sentences seem not relevant to the findings of this study. Please discuss the present results by referring to earlier studies on herbivory and detritivory rates as well as ecosystem properties associated with P cycles in Amazon and other ecosystems. To do this, some sentences could be brought from Introduction section.

The introduction, methods, results and discussion sections were re-structured to make clear that this paper aims to make a synthesis of the Amazon basin P cycle. Our study in particularly explores and provides insights about the importance of animal redistribution of P within and between seasonally flooded and non flooded ecosystems (within each sub-basin). Although the paper became more extensive, we hope this new structure makes it easier for readers to follow and interpret our results. Minor comments: Materials and Methods P8L8(Fig.3): If P transport from flooded area to non-flooded area (terra firme) is not modeled in this study, it would be better to depict the two boxes representing herbivores or detritivores separately for each area.

We have modified the figure 1 to make clear that we consider herbivory and detritivory fluxes within the sub/basin (the first mode of P transport mentioned in a reply above); we have also corrected the names of some of the variable in figure 3 better specifying the in- and out-fluxes. We hope now it is clear that we are actually modelling the animal fluxes between seasonally flooded and terra firme ecosystems within the same sub-basin.

P8L15: Please explain the equation and each symbol more carefully here. What OoE stands for?

Thanks for pointing this out, we have now re-written the description of the parameters, and checked the consistency of the symbols, text, tables and in fig. 3.

P10L3: discussed

Changed

P10L10-16: These sentences have been described already in Introduction, and thus could be deleted.

This paragraph has been deleted

P10L17: It would be needed to explain about Rio Negro basin and its relation to the overall Amazon here.

We expanded the explanation to:

"Rio Negro is a sub-basin draining only the lowland of the Amazon basin and therefore the weathering rates measured can be taken as a good proxy for lowland amazonian weathering"

P10L21: I could not understand how the value 1895.84 molP per km2 per year was calculated. Please explain.

More explanation was added to the MS "Redistributing all the P that is weathered in the Andes to the whole Amazon basin results in 246 molP in XX per km2 (of whole Amazon basin) per year, the area where this comes occupies only 13% of the total area. We divide the previous number by 0.13 to get the in-situ weathering, getting then 1895.84 molP per km2 (Andean part) per year. All chemical weathering does not occur in-situ, but also during the transport and after the deposition in the river and seasonally flooded areas".

P11L3: A reference should be needed here.

We added some more information to explain why the sensitivities For the animal P flux from rivers to the flooded areas, simulations with three different P inputs were run, with values of 0, 72 and 242 g P ha−1 a−1 . The first one simulates a scenario with no animals, the second simulates a

270 scenario in which P transfer is like the one is Suriname due to giant otter (Pteronura brasiliensis), with an associated P flux of 72 g P ha−1 a−1 as it was estimated in the introduction, and the last one simulates a scenario in which otters and other animals contribute, since this contribution is totally unknown, the limit for the sensitivity analysis was set to a value between 3 to 4 times of the previous one, 242 g P ha−1a−1.

P11L4-11: These sentences seem to have been explained already, and thus redun-Dant.

The sentence was removed.

Results

P12L6: What 4 vs.5 stands for?

We have corrected this issue.

P12L16: From where is P transported to terra firme ecosystem? If it comes from terra firme vegetation itself, the expression "transport" should be misleading. The model redistributes the P transport from one part of the ecosystem to others and also between ecosystems.

We have added a new figure, now figure 4, in which the net animal fluxes to terra firme ecosystems are simulated in relation to herbivory and detritivory.

P13 Figure.4: Please change the order of the panels so as to match the order of site explanations in text (1, Rio Negro, 2, Caqueta-Japura, 3 Cerrado). And, add (%) to the x axis label as in Figure 6.

We changed the scale to % and change the order given in the explanation, now we are also referring to Xingu sub-basin instead of Cerrado to be more consiste with language and avoid confusion.

P13L3: delete "we"

Changed

P14L5: What does "s<0.4" mean?

Given our model parametrization dry ecosystems are those that have a relative volumetric soil moisture content (denoted by "s") less than 0.4. We erased the value in parenthesis to avoid confusion.

Discussion P15L8: Starting new paragraph here should not be necessary.

Changed

P15L9: different results between what?

The phrase was changed to: Second, animal driven redistribution leads to contrasting P in vegetation depending on herbivore grazing pressure

P17L2: redistribution, for example, .... The phrase was changed to: It would be interesting to assess if other models of nutrient redistribution exhibit a similar transition from positive to negative effects (for example, in the case of mega-herbivores before the megafauna extinction, see \citet{Doughty:2013})

P17L16: Again, it is unclear from where the animals transport P into terra firme ecosystems. Please explain.

P redistribution by animals represents a net P flux from seasonally flooded to terra firme ecosystems. The text has also been modified to: This is consistent with our model finding that redistribution is key to the P budget of the \emph{terra firme} ecosystems and that detritivores are particularly important in this process (fig. \ref{fluxes} and \ref{states})

P17L17-18:This sentence could be deleted.

Deleted

P17L19-P18L6: These sentences could be deleted, because they are not directly rel-

evant to the present results, and seem very speculative.

Now the manuscript has been re-structured to differentiate between a specific discussion of model results and a more general discussion synthesizing the P cycling of the Amazon basin as a whole. Subsections in the introduction as well as in the discussion section have been included to make this distinction clearer .

P18L6-7: The sentence seems lack of scientific basis.

It is widely recognized (also in the scientific community) that traditional knowledge is highly valuable (make reference to IPBES or Aichi targets of the CBD). In this case, we would like to emphasize that complex food-webs well-known to local people but perhaps less understood or studied by the scientific community can play a major and possibly under-appreciated role in driving P cycling.

P18L14-P19L17: These sentences concerning human impacts and megafauna should not be the main topics of this manuscript, and therefore could be deleted

Even though these topics are not the main objective of the paper, we consider them important to mention, especially in the context of a broader and more holistic discussion about Amazon P cycling, which is now explicit in the discussion section. We argue that even without the action of megaherbivores (Doughty et al. 2013), P can effectively redistributed across Amazon landscapes. Even more, , during pre-colombian times humans have invented management strategies that modified the P biogeochemical cycle of the Amazon, also in part against the physical flow gradient e.g. P-enriched terra preta soils in terra firme ecosystems limiting P leaching. In the context of future applications of our model, it may be worth mentioning these points .

Please also note the supplement to this comment:
https://www.biogeosciences-discuss.net/bg-2017-121/bg-2017-121-AC2-supplement.pdf

---

## Author Response (AR1)

Dear Editor,

We thank you very much for your patience. We have responded carefully to the comments by both reviewers and made a great effort to restructure the manuscript, improve its coherence and thus make it easier to understand for the reader.

The introduction, methods, results and discussion sections were re-structured to make clear that this paper aims to make a synthesis of the P cycle of the Amazon basin  In particular, using a simple mathematical model our study explores and provides insights about the importance of P redistribution by animals  within and between seasonally flooded and non flooded ecosystems in the context of 3 different sub-basin Although the paper became more extensive, we hope this new structure makes it easier for readers to follow and interpret our results.

One of the main concerns by Reviewer 1 was a possible error in one of our equations. In the reply and in the revised manuscript we explained the logic behind the equations, provide some additional plots that were suggested so that it becomes easier for the reviewer to check that the model is correct.  The new suggested plots are now included in the manuscript as we found them useful to make he model results easier to followed.

Another substantial change is adjusting the names of the sub-basins. We now refer to Cerrado as Xingu sub basin since the Cerrados are not a sub-basin in the strict sense.

In the new version of the manuscript we took into consideration all comments and suggestions by Reviewer 2 and hope to have addressed them satisfactory. Specifically, we extensively re-worked the results section to explain better model outputs and results, and more clearly separated discussion points stemming from the model results from those stemming from the synthesis of previous works on P cycling, explained them in more detail in the context of the Amazon basin.

Best regards,

Corina on behalf of all co authors